# On Share Conversions for Private Information Retrieval

**DOI:** 10.3390/e21090826

**Published:** 2019-08-23

**Authors:** Anat Paskin-Cherniavsky, Leora Schmerler

**Affiliations:** Computer Science Department, Ariel University, Ariel 40700, Israel

**Keywords:** PIR, share conversion, CNF secret sharing, communication complexity

## Abstract

Beimel et al. in CCC 12’ put forward a paradigm for constructing Private Information Retrieval (PIR) schemes, capturing several previous constructions for k≥3 servers. A key component in the paradigm, applicable to three-server PIR, is a share conversion scheme from corresponding linear three-party secret sharing schemes with respect to a certain type of “modified universal” relation. In a useful particular instantiation of the paradigm, they used a share conversion from (2,3)-CNF over $Z_{m}$ to three-additive sharing over $Z_{p}^{\beta }$ for primes $p_{1},p_{2},p$ where $p_{1}\neq p_{2}$ and $m\;=\;p_{1}\cdot p_{2}$. The share conversion is with respect to the modified universal relation $C_{S_{m}}$. They reduced the question of whether a suitable share conversion exists for a triple $\left(p_{1},p_{2},p\right)$ to the (in)solvability of a certain linear system over $Z_{p}$. Assuming a solution exists, they also provided a efficient (in m,logp) construction of such a sharing scheme. They proved a suitable conversion exists for several triples of small numbers using a computer program; in particular, $p\;=\;p_{1}\;=\;2,p_{2}\;=\;3$ yielded the three-server PIR with the best communication complexity at the time. This approach quickly becomes infeasible as the resulting matrix is of size $\Theta (m^{4})$. In this work, we prove that the solvability condition holds for an infinite family of $\left(p_{1},p_{2},p\right)$’s, answering an open question of Beimel et al. Concretely, we prove that if $p_{1},p_{2}>2$ and $p\;=\;p_{1}$, then a conversion of the required form exists. We leave the full characterization of such triples, with potential applications to PIR complexity, to future work. Although larger (particularly with $max(p_{1},p_{2})>3$) triples do not yield improved three-server PIR communication complexity via BIKO’s construction, a richer family of PIR protocols we obtain by plugging in our share conversions might have useful properties for other applications. Moreover, we hope that the analytic techniques for understanding the relevant matrices we developed would help to understand whether share conversion as above for $C_{S_{m}}$, where *m* is a product of more than two (say three) distinct primes, exists. The general BIKO paradigm generalizes to work for such $Z_{m}$’s. Furthermore, the linear condition in Beimel et al. generalizes to *m*’s, which are products of more than two primes, so our hope is somewhat justified. In case such a conversion does exist, plugging it into BIKO’s construction would lead to major improvement to the state of the art of three-server PIR communication complexity (reducing Communication Complexity (CC) in correspondence with certain matching vector families).

## 1. Introduction

A Private Information Retrieval (PIR) protocol [1] is a protocol that allows a client to retrieve the ith bit in a database, which is held by two or more servers, each holding a copy of the database, without exposing information about *i* to any single server (assuming the servers do not collaborate). In the protocol specification, the servers do not communication amongst each other. The main complexity measure to optimize in this setting is the communication complexity between client ans servers. In the single-server setting, the Communication Complexity (CC) is provably very high - provably, the whole database needs to be communicated. In the computational setting [2], communication-efficient single-server PIR is essentially solved with essentially optimal CC [3]. In this work, we focus on information theoretic (in fact perfect) PIR protocols. See [4] and the references within for the additional motivation for the study of information theoretic PIR.

All known PIR protocols only use one round of communication (although it is not part of the definition of PIR), so we will only consider this setting. In this setting, the client sends a query to each server and receives an answer in return. PIR is a special case of secure Multi-Party Computation (MPC), which is a very general and useful cryptographic primitive, allowing a number of parties to compute a function *f* over their inputs while keeping that input private from an adversary that may corrupt certain subsets of the parties (to the extent allowed by knowing the output of *f*) [5,6]. The PIR setting is useful on its own right, as a minimalistic useful client-server setting of MPC, where the goal is to minimize communication complexity, potentially minimizing the overhead on the client, which may be much weaker than the server.

In [4], the authors proposed the following approach to constructing PIR protocols, which captures some of the previous protocols for three servers or more, and put forward a new three-server PIR protocol, with the best known asymptotic communication complexity to date.

Let us describe their general framework, for a *k*-server PIR.

### 1.1. Biko’s Framework for *K*-Server PIR

The framework uses two key building blocks. One is a pair of *k*-party linear secret sharing schemes $Sh_{1},Sh_{2}$ over abelian rings $G_{1},G_{2}$, respectively. The pair $Sh_{1},Sh_{2}$ is also equipped with a share conversion scheme from $Sh_{1},Sh_{2}$ with respect to some “useful” relation $R\subseteq G_{1}\times G_{2}$. That is, for a value *s* shared according to $Sh_{1}$, the scheme allows locally (performing a computation on each share separately) computing a sharing of a value *o* according to $Sh_{2}$, such that R(s,o) holds (note that this is generally non-trivial, as the conversion is performed locally on each share, without knowing anything about the other shares or the randomness used to generate the sharing).

In [4], $G_{1},G_{2}$ used small (constant) finite rings. The notion of share conversion employed in [4] generalized [7]’s work on locally converting from the arbitrary linear secret-sharing schemes without changing the secret, by allowing an arbitrary relation between the secrets and by allowing $Sh_{1},Sh_{2}$ to be linear over different rings. The second building block is an encoding of the inputs $x\in {\left\{0,1\right\}}^{n}$ as a longer element $u\in G_{1}^{l}$. The PIR protocol has the following structure:The client “encodes” the input $x\in {\left\{0,1\right\}}^{n}$ via $u\in G_{1}^{l}$ for a quite large, but not very large l(n) from a certain code $C\subseteq G_{1}^{l}$.Every bit $u_{j}$ is shared via $Sh_{1}$. The client sends each share $s_{i,j}$ to server $S_{j}$.The servers are able to evaluate locally any shallow circuit from a certain set F roughly as follows. At the bottom level, we use the linearity of $Sh_{1}$ to evaluate a sequence of gates taking linear combinations of the $u_{i}$’s over $G_{1}$. Let *y* denote the resulting shared values vector. Apply the share conversion to the output of each of the $y_{i}$’s to obtain a converted vector $y^{\prime }$ of values over $G_{2}$. Finally, locally evaluate some linear combination of the elements $y^{\prime }$, using the linearity of $Sh_{2}$. Each server sends its resulting share to the client. The particular circuit to evaluate is that of an “encoding” $f^{\prime }:C\to G_{2}$ of their database, in turn viewed as a function $f:{\left\{0,1\right\}}^{n}\to \left\{0,1\right\}$.The client reconstructs the output value from the obtained shares, using $Sh_{2}$’s reconstruction procedure. It then decodes f(x) from $f^{\prime }\left(u\right)$ (the decoding procedure takes only $f^{\prime }\left(u\right)$ as input; it does not use *x* or the client’s randomness).

Note that the “privacy” property of $Sh_{1}$ takes care of keeping the client’s input private from any single server. It is not known how, and likely impossible, to devise a share conversion scheme for a relation that is sufficiently useful to compute locally all functions $f:{\left\{0,1\right\}}_{1}^{l}\to \left\{0,1\right\}$ following the framework suggested above, in a way that encodes every $f:{\left\{0,1\right\}}^{l}\to \left\{0,1\right\}$. Therefore, to encode all functions $f:{\left\{0,1\right\}}^{l}\to \left\{0,1\right\}$, BIKOencodes the inputs *x* and function $f:{\left\{0,1\right\}}^{n}\to \left\{0,1\right\}$ using a larger parameter l>n. To gain in communication complexity, the code C is carefully chosen so that the resulting family $F\subseteq \{f^{\prime }:G_{1}^{l}\to \left\{0,1\right\}\}$ implemented by the above protocol is as large as possible (relative to *l*). More precisely, its VCdimension is at least $2^{n}$, allowing one to implement any Boolean function on ${\left\{0,1\right\}}^{n}$. On the other hand, we must be able to match it with a relation *R* for which a share conversion exists (Recall that the VC dimension captures the ability of a set F of functions f:D→{0,1} to “encode” any function $f:{\left\{0,1\right\}}^{n}\to \left\{0,1\right\}$ by restriction to a subset I⊆D of size $2^{n}$ of values. Here, *D* is typically larger than ${\left\{0,1\right\}}^{n}$. By Saur’s lemma, the VC dimension is very closely related to the size of the family F.) There is a tradeoff between the share conversions we are able to devise and the complexity of F. The larger F in the above scheme is, the smaller *l* that can be used for the encoding, and the lower the communication complexity of the resulting PIR protocol. Note that the framework yields constructions where the communication complexity of O(l) bits is dominated by the client’s message length and only constant-length server replies. On the other hand, for more complex F, one would need share conversion schemes for “trickier” relations, which are harder to come by.

Only a few families of instantiations of the framework are known, with [4]’s particular instantiation for three-server PIR as one example. Other instantiations are implicit in some previous PIR protocols; see Section 2, or [4] for additional examples and more intuition.

#### 1.1.1. BIKO’s New Family of Three-Server PIR

In their new three-server PIR, the work in [4] instantiated the components of the above framework using a certain (family of) rings $G_{1},G_{2}$, for which suitable encodings and share conversion exist. Details follow. The rings are parameterized by three numbers $p_{1},p_{2},p$, where $p_{1},p_{2}$ are distinct primes. Let us denote $m=p_{1}\cdot p_{1}$. The various components are instantiated as follows:$Sh_{1}$ is the (2,3)-CNF secret sharing scheme for the ring $G_{1}=Z_{m}$, for certain $m=p_{1}\cdot p_{2}$, where $p_{1},p_{2}$ are distinct primes. $Sh_{2}$ is the three-additive secret sharing over the ring $G_{2}=Z_{p}^{\beta }$, for some β∈N.Let $S\subseteq Z_{m}\backslash \left\{0\right\}$. The input $x\in {\left\{0,1\right\}}^{n}$ is encoded as an element *u* of an *S*-Matching Vector (MV) $C\subseteq Z_{m}^{l}$ family of size at least $2^{n}$ over $Z_{m}$ [8]. Briefly, for $S\subseteq Z_{m}\backslash \left\{0\right\}$, such an *S*-MV family is a sequence of vectors $v_{1},\ldots ,v_{L}\in Z_{m}^{l}$, where for all *i*, $<v_{i},v_{i}>=0$, and for all i≠j, $<v_{i},v_{j}>\in S$. In particular, [8] demonstrates that an *S*-MV family of any VC dimension exists for all $m=p_{1}p_{2}$ where $p_{1},p_{2}$ are distinct primes for *S* as small as $S_{m}$; the set of all elements in $Z_{m}$ that are zero or one modulo each of the $p_{i}$’s, expect for zero. In particular, the vector’s length for an instance corresponding to VC dimension $2^{n}$ is $l=2^{\tilde{O}\left(\sqrt{n}\right)}$.A share conversion between $Sh_{1}$ and $Sh_{2}$ for a relation $C_{S_{m}}$ is obtained for certain triples $\left(p,p_{1},p_{2}\right)$. Informally, this relation maps $S_{m}$ to zero and zero to some non-zero value in $G_{2}$. This type of share conversion scheme is another novel technical contribution of [4] (and the focus of our work).

The complete BIKO construction is roughly as follows:The client sends an encoding u=u(x) of its input $x\in {\left\{0,1\right\}}^{n}$, which belongs to the the $S_{m}$ MV-family $C\subseteq Z_{m}^{l}$ (here, $l=O(2^{\sqrt{nlog(n)}})$ can be achieved). The servers locally evaluate the following circuit:
$$C\left(u_{1},\ldots ,u_{l}\right)=OR_{j}\left(f_{i}\cdot <u_{j},u>\right)$$
using the following “noisy” procedure:Generate the vector of values ${\left(y_{i}\right)}_{f(i)=1}$ where $y_{i}=<u_{i},u>$. This evaluation uses the linearity of CNF and produces a local $Sh_{1}=\left(2,3\right)$-CNF over the $Z_{m}$ share of each value $y_{i}$.Apply share conversion from $Sh_{1}$ to $Sh_{2}$ as specified above on each $y_{i}$, obtaining a vector of $y_{i}^{\prime }$’s, shared according to $Sh_{2}$.Locally evaluate $\sum_{i}y_{i}^{\prime }$, and then, send ($Sh_{2}$) shares to the client. The client outputs zero if this value is zero, and one otherwise.

It is not hard to see that the above construction always produces the correct value. In particular, by the definition of $C_{S_{m}}$, only (the conversion of) $<u_{i},u_{i}>$ contributes a non-zero value $y_{i}^{\prime }$ to $\sum_{i}y_{i}^{\prime }$.

**Theorem** **1** ([4], informal)**.**
*Assume there exists a share conversion scheme from (2,3)-CNF over $G_{1}=Z_{m}$ and three-additive over $G_{2}=Z_{p}^{\beta }$ for some β, where $m=p_{1}\cdot p_{2}$ is as above. Then, there exists a three-server PIR with communication complexity $O(2^{2\cdot p_{2}\sqrt{nlog(n)}})$, where $p_{2}>p_{1}$. The best asymptotic communication complexity is obtained by setting m=6,p=2, using [8]’s $S_{m}$-MV family over $G_{1}^{l}$ over $Z_{6}$ with $l=2^{\tilde{O}\left(\sqrt{n}\right)}$ and a share conversion with respect to $C_{S_{m}}$ for $p_{1}=p=\beta =2,p_{2}=3$. In particular, the servers’ messages are of length O(1).*

Intuitively, having *S*-MV families for a “small” *S* as above facilitates the construction of share conversion schemes as we needed for $C_{S}$, since a small *S* imposes fewer constraints on the pairs $\left(s,s^{\prime }\right)$ in the relation.

To conclude, it follows from Theorem 1 that to obtain three-server PIR with sub-polynomial (in database size) CC, it suffices to design a share conversion for $C_{S_{m}}$ from $Sh_{1}$ to $Sh_{2}$ over groups $G_{1},G_{2}$ corresponding to any triple of parameters $p_{1},p_{2},p$ as described above. Another useful contribution of [4] is reducing the question of whether such a conversion exists (for some β) to a system of equations and inequalities over $F_{p}$; see the following sections for more details on the usefulness of this characterization (a solution to the system corresponds to the share conversion scheme).

#### 1.1.2. On the Choice of Input and Output Schemes

In retrospect, trying to convert from CNF to additive is a natural choice. For once, the CNF scheme is known as the most redundant secret sharing scheme and additive is the least redundant one [7]. More specifically, CNF sharing is convertible to any other linear scheme over the same field for the same access structure, or one contained it with respect to the identity relation. This implies that for any relation, if share conversion between two linear schemes exists for that relation, it must exist for CNF to additive. Here the more standard linearity over fields is considered, and the same field us used for both schemes. Still, for the more general notion of linearity over rings, and allowing different rings for the two schemes, this still provides quite strong evidence that starting with CNF, and trying to convert to additive is a good starting point. As to the choice of the structure of the particular $G_{1},G_{2}$ over which the conversion is defined, following are some of [4]’s considerations for their choice. One observation is that trying to convert from, say, $Z_{m}$ to $Z_{p}$ would require finding a solution to a system of inequalities over $Z_{p}$ for a fixed *p*. This problem is generally NP-complete, which is likely to make the problem hard to analyze, even in this special case. Thus, the authors extend the search to allow conversion to three-additive over some extension field $F_{p^{\beta }}$. As β is constant, it does not have much effect on the share complexity of the resulting PIR protocol.

Using composite sizes for both $G_{1}=Z_{m}$ and $G_{2}=Z_{m^{\prime }}$ would lead to a greatly improved VC dimension of F, with respect to the canonical relation $C_{S}$, where $S=Z_{m}\backslash \left\{0\right\}$. In fact, the $F={+}_{Z_{m^{\prime }}}\left(mod_{m}\right)$ that can be locally evaluated over $Z_{m}^{l}$ is universal, resulting in near-optimal O(log(n)) communication complexity. However, [4] have proven this type of conversion does not exist, indicating that $C_{S}$ for smaller *S* for such $G_{1},G_{2}$ could also be hard to find.

### 1.2. Our Contribution

As described above, the contribution of [4] was two-fold. First, they put forward a useful framework for designing PIR protocols, capturing some of the best-known three-server (or more) PIR protocols. Second, they put forward an instantiation of the framework, which reduces the question of the existence of a three-server PIR protocol to the existence of a share conversion for certain parameters $p_{1},p_{2},p$ and certain linear sharing schemes over abelian rings $G_{1},G_{2}$ determined by the parameters. They further reduced the question of the existence of a share conversion as above for parameters $p_{1},p_{2},p$ to the question of whether a corresponding system of linear equations is solvable over $Z_{p}$. This system in turn is not solvable over some extension field $F_{p^{\beta }}$ iff a certain system of both equalities and inequalities over $Z_{p}$ is solvable over some field $F_{p^{\beta }}$.

While the solvability of a system Ax=b can be verified efficiently for a concrete instance, it does not provide a simple condition for characterizing triples $\left(p,p_{1},p_{2}\right)$ for which solutions exist. More concretely, even the question of whether an infinite set of such triples exists remains open. We resolve this question in the affirmative, proving the following.

**Theorem** **2.**
*Let $2<p_{1},p_{2}$ where $p_{1}\neq p_{2}$ are primes. Then, a share conversion from (2,3)-CNF over $Z_{p_{1}p_{2}}$ to three-additive over $Z_{p_{1}}^{\beta }$ exists for some β>0.*


In a nutshell, our goal is to study the solvability of the particular system Ax=b from [4] corresponding to a given $p,p_{1},p_{2}$. Our main tool is a clever form of elimination operations on the (particular) matrix (A;b), so that at each step, the intermediate system $A^{\prime }x=b^{\prime }$ is solvable iff Ax=b is. The special type of elimination we develop is useful here as it allows keeping our particular system relatively simple at every step. This is as opposed to using standard Gaussian elimination, for instance, which would have made the resulting system quite messy. The operations are largely oblivious of the particular value of $p_{1},p_{2},p$ until a very late point in the game. The proof consists of a series of a (small) number of our elimination steps. On a high level, we perform a few such steps. At this point, the matrix becomes quite complicated (at the end of Section 4.2). Then, we perform a change of basis (Section 4.5) to facilitate another convenient application of the step. At this point, we are ready to complete the proof by directly proving that the system is not solvable (and thus, a share conversion exists).

#### 1.2.1. On the Potential Usefulness of Our Result for PIR

In this work, we identified a certain infinite set of parameters $\left(p_{1},p_{2},p\right)$ for which a share conversion as required in Theorem 1 exists. This result itself does not appear to yield improved three-server PIR protocols by instantiating the share conversion with the newly-found parameters via Theorem 1. Indeed, increasing $p_{2}$ beyond the minimal possible $p_{2}=3$ (which was already known) does not seem to help. However, Theorem 1 generalizes to *m*’s, which are products of a larger number of primes. If a share conversion for $G_{1}=Z_{m},G_{2}=Z_{p}^{\beta }$ derived from such $m=p_{1}\cdot p_{2}\cdot p_{3},p$ exists, it can be paired with an $S_{m}$-MV family (which is also known to exist from [8]) with significantly improved (over *m* a product of two primes) VC dimension, to obtain an improved three-server PIR seamlessly. Specifically for $m=p_{1}\cdot \ldots \cdot p_{r}$, an $S_{m}$ MV family with VC dimension $2^{n}$ in $Z_{m}^{l}$ for $exp(\tilde{O}\left(log^{1/r}\left(n\right)\right))$ exists.

Therefore, it remains to design a corresponding share conversion scheme (or prove it impossible to rule out this direction). Furthermore, the linear algebraic characterization of the existence of such sharing schemes remains the same. Therefore, hopefully, the analytic techniques we developed here for the the case where *m* is a product of two primes that could help understand the case of a larger number of primes.

#### 1.2.2. Road Map

In Section 2, we refer to some related work on PIR. In Section 3, we include some of the terminology and preliminary results from [4], which is our starting point. In Section 4, we present our main result, which is broken into subsections as follows. As explained above, we start with the reduction of the problem by [4] to verify whether a certain matrix-vector pair (A,b), Rows(A) spans *b*. We perform a series of elimination steps on the matrix (A|b) to bring it to a simpler form $\left(A^{\prime },b^{\prime }\right)$ so that $Rows(A^{\prime })$ spans $b^{\prime }$ iff Rows(A) spans *b*. All elimination steps are of the same general form. Section 4.1 formalizes this form as a certain lemma. Section 4.2 outlines an initial sequence of applications of that lemma. In Section 4.5, we perform a change of basis to help facilitate further use of the lemma. In Section 4.6, we indeed apply the lemma once again. Finally, in Section 4.7, we obtain a simple enough matrix $\left(A^{\prime }|b^{\prime }\right)$ for which we are able to prove $A^{\prime }$ spans $b^{\prime }$ directly for the proper choice of parameters, proving our positive result.

## 2. Related Work

### 2.1. Schemes with Polynomial CC

As stated in the Introduction, our work is a followup on a particular approach to constructing PIR protocols [1], focusing on k=3 servers (almost the hardest case). We will survey some of the most relevant prior works on PIR (omitting numerous others). Already in [1], the authors suggested a non-trivial solution to the hardest two-server case, with communication complexity $O(2^{n/3})$, and more generally, $O(2^{\frac{n}{k}})$ for k≥3 servers. Building on a series of prior improvements, the work in [9] put forward protocols with CC $2^{\frac{O(n\cdot loglogk}{klogk})}$ for large *k*. Their result also improved upon the state-of-the-art at the time for small values of *k*, in particular achieving CC of $O(2^{4n/21})$ for k=3, improving upon the best previously-known CC of $O(2^{n/5})$.

#### Work Falling in the Framework on BIKO

In [9], they restated some of the previous results in a more arithmetic language, in terms of polynomials. Furthermore, they considered a certain encoding of the inputs and element-wise secret sharing the encoding, which is somewhat close to the BIKO framework.

Particularly interesting for our purposes is their presentation of a “toy” protocol achieving CC of $O(2^{n/2})$ from earlier literature reformulated in [9], which constituted one of the building blocks of the construction of [9]. That protocol was almost an instance of the BIKO framework as sketched in the Introduction, and it is instructive to consider the differences (the full-fledged result of [9] used additional ideas, and we will not go over it here for the sake of simplicity).
The client “encodes” its input $x\in {\left\{0,1\right\}}^{n}$ via $u\in F_{2}^{l}$ where $l\approx 2^{n/2}$ by mapping the input to a vector with exactly two ones and zeroes elsewhere.Every bit $u_{j}$ is shared via $Sh_{1}$, which is (2,3)-CNF over $F_{2}$.The servers represent their database as a degree-two polynomial in $F[x_{1},\ldots ,x_{l}]$ (where all monomials are of degree exactly two). To evaluate a polynomial:
$$f\left(x\right)=\sum_{i<j\leq l}a_{i,j}x_{i}x_{j}$$
proceed as follows:
(a)At the bottom level, perform a two-to-one share conversion for evaluating each monomial, where the output for gate $x_{i}x_{j}$ is a three-additive sharing of the result $x_{i}x_{j}$. In some more detail, let $x_{i,1}+x_{i,2}+x_{i,3}$ denote the additive shares produced in Step 1 of the CNF-sharing. The computation is made possible since $\left(x_{i,1}+x_{i,2}+x_{i,3}\right)\cdot \left(x_{j,1}+x_{j,2}+x_{j,3}\right)$, which needs to be computed; for every share-monomial $x_{i,k}x_{k,d}$, at least one of the servers does not miss any shares in it. Each server outputs the sum of the monomials it knows as its new share.(b)Using the linearity of the three-additive scheme, the servers compute a three-additive sharing f(x), based on the shares of the individual monomial evaluations $y_{i}$.Each server sends its resulting share to the client.The client reconstructs the output value from the obtained shares, as in BIKO.

This framework differs from BIKO in a few ways. First, the structure of the circuit locally evaluated is different. In the above example, “many-to-one” rather than “one-to-one” secret sharing is used at the bottom level. Then, the linearity of $Sh_{2}$ is employed on the upper level. In BIKO, first, the linearity of $Sh_{1}$ is used to evaluate a vector of linear combinations, then a one-to-one share conversion from $Sh_{1}$ is used at the middle level, then again, the linearity of $Sh_{2}$ is used at the top gate. Besides the different structure of the circuit, using many-to-one share conversion is the main conceptual change. It employs the extra property of (2,3)-CNF, which allows it to evaluate degree-two polynomials, rather than just linear functions. In the evaluation process, the output is already converted to the less redundant three-additive sharing. The relation the share conversion respects is just the evaluation of the monomial: $R(o,x_{1},x_{2})$ iff $o=x_{1}x_{2}$. Conceptually, local evaluation of linear functions as used in BIKO is in fact already a many-to-one share conversion from a linear scheme to itself, evaluating a relation $R(x_{1},\ldots ,x_{l},o)$, which is satisfied iff $o=\ell (x_{1},\ldots ,x_{l})$ for a certain linear function *ℓ*.

### 2.2. Schemes with Sub-Polynomial CC

The first three-server PIR protocol with (unconditionally) sub-polynomial (in $2^{n}$) CC was put forward by [10]. It falls precisely into BIKO’s framework, using a share conversion from the so-called (2,3)-modified Shamir secret over $G_{1}=Z_{m}$ for $m=p_{1}\cdot p_{2}$ where $p_{1}\neq p_{2}$ are odd primes to three-additive over the field $G_{2}=F_{p^{\beta }}$ where *p* is some prime, such that *m* divides $|G_{2}^{*}|=p^{\beta }-1$.

To get to three, rather than four servers, some additional properties of $G_{1},G_{2}$ were in fact required. The share conversion is with respect to the relation $C_{S_{m}}$ (actually, a slightly stricter relation, even). Concretely, the work in [10] found an example of such groups $G_{1},G_{2}$ as above obtained from m=7·73=511, p=2,β=9, having an additional useful property, which allowed going down from four to three servers via computer-aided search. We note that a share conversion from (2,3)-CNF to three-additive over that pair of groups also exists since (2,3)-CNF can be converted to any linear scheme for an access structure containing a two-threshold (including modified (2,3)-Shamir) [7].

The encoding used is via a $S_{m}$-MV family over $Z_{m}$, as in [4]. The evaluation points in the Shamir scheme used are tailored to $S_{m}$ in the corresponding MV family. It achieves a CC of $O(2^{146\sqrt{n\log(n)}})$, later improved to $O(2^{6\sqrt{n\log(n)}})$ by BIKO’s construction.

Qualitatively, the result of [10], preceded by [11] in terms of using a similar idea, and most later constructions greatly improved the CC relative to earlier constructions such as [9] by using MV-codes rather than low-degree polynomial codes. The former codes had a surprisingly large rate. Indeed, looking at MV-families from the perspective of [12], these are also polynomial codes, over a basis of monomials corresponding to an MV family, rather than low-degree monomials.

All the above examples generalize to larger *k*, with somewhat improved complexity; we focused on the (hardest) k=3 here for simplicity.


*Two-Server PIR*


In a breakthrough result, the work in [13] matched the CC of two-server PIR with the CC of the best-known three-server PIR, improving from the best previously-known $2^{n/3}$ to $2^{O(\sqrt{nlog(n)})}$. Their work cleverly combined and extended several non-trivial ideas, both new and ones that appeared in previous work in some form. In a nutshell, one idea is to encode inputs via MV codes, where the operations on the vectors are done “in the exponent”, resulting in a variant of MV codes as polynomials, as defined in [12] and implicitly used in [10]. Another idea is to output both evaluations of these “exponentiated” polynomials and the vector of their partial derivatives, to yield more information to facilitate reconstruction given only two servers.

As for casting their result into the BIKO framework(we believe it is a meaningful generalization thereof, as we describe below):The client encodes its input $x\in {\left\{0,1\right\}}^{n}$ via a vector u=u(x) in an $S_{6}$-MV family $C\subseteq Z_{6}^{l}$ (with *l* about $2^{\tilde{O}\left(\sqrt{n}\right)}$ as explained above).Every bit $u_{j}$ is shared via $Sh_{1}$, which is the (2,2)-modified Shamir scheme over $Z_{m}$. The evaluation points are zero and one. We note that the choice of *m* is not as constrained as in the construction of [10], and many other pairs would work as well. This holds as the purpose of the polynomial involved is somewhat different.Let $R_{6,6}$ denote the ring $Z_{6}\left[z\right]/\left(z^{6}-1\right)$. Let γ=z, which has multiplicative degree six in $R_{6,6}$. The servers are able to evaluate locally the encoded database:
$$F\left(x_{1},\ldots ,x_{l}\right)=\sum_{i\in [2^{n}]}a_{i}x^{u_{i}}$$
roughly as follows:
(a)At the bottom level, use the linearity of $Sh_{1}$ to evaluate locally each $y_{i}=<u_{i},x>$. (b)Convert each of the $y_{i}$’s into a (2,2)-sharing scheme $Sh_{2}$ over a ring $G_{2}=R_{6,6}^{\Omega (l)}$, obtaining a vector of $y_{i}^{\prime }$’s. The conversion is a many-to-one conversion with respect to a relation *R* over the entire input vector, the current $y_{i}$ and $y_{i}^{\prime }$, so that $R(x_{1},\ldots ,x_{2^{n}},y_{i},y_{i}^{\prime })$ that outputs $C_{S_{6}}\left(y_{i},y_{i}^{\prime }\right)$ if $\left(x_{1},\ldots ,x_{2^{n}}\right)$ is in C, and otherwise undefined.(c)Use the “almost linearity” of $Sh_{2}$ over our particular subset of values to evaluate $\sum_{i}a_{i}y_{i}^{\prime }$. Here. “almost linearity” means that to evaluate a sum of shared secrets, we add some of the coordinates and copy other coordinates.The client reconstructs the output value from the obtained shares, according to $Sh_{2}$ (the reconstruction function is of degree-two.) In a nutshell, the coefficient $a_{x}$ is the only one determining the free coefficient of a degree-six univariate polynomial obtained from restricting *F* to the line z+ut, where *z* is a random vector. The reconstruction procedure uses the shared evaluations as derivatives to compute this free coefficient, which is non-zero iff $a_{x}$ is one. In the original protocol description of [13], the client used the input for reconstructing the output, which is not consistent with share conversion, where the input is not required for reconstruction. We restate the above protocol by letting the servers “play back” the original shares to the client to stay consistent with the BIKO share conversion framework.

To summarize, the construction falls into an extended BIKO framework, where the middle-layer share conversions are many-to-one rather than one-to-one. Furthermore, the scheme $Sh_{2}$ is non-linear and is defined over a non-constant domain. It is important to observe that the relations used for the conversions are independent of the database function *F* itself, but only depend on its size. The locally-evaluated relations are as small as $O(2^{n})$, similarly to BIKO. One potential difficulty with using such an extended design framework is in verifying whether a many-to-one share conversion exists, or even one-to-one for a large share domain of $Sh_{2}$, as we have here.

One question that remains open in the two-server setting is to make the server’s output size constant as in [4,10] for three servers.

## 3. Preliminaries

### 3.1. Secret Sharing Schemes

For a set A⊆[n] and sequence of shares $s=(s_{1},\ldots ,s_{n})$, we denote s[A] by the sequence of shares ${\left(s_{i}\right)}_{i\in A}$. A secret sharing scheme for *n* parties implements an access structure specified by a monotone function $f:{\left\{0,1\right\}}^{n}\to \left\{0,1\right\}$ specifying the so-called *qualified* sets of parties that recover the secret (preimages of one), while other sets learn nothing about the secret (in this paper, we consider the standard perfect setting). We refer to such a scheme as an *n*-party secret sharing scheme. More formally, an *n*-party secret sharing scheme is a randomized mapping:$$Sh:S\times R\to S_{1}\times \ldots \times S_{n}$$
where *S* is a finite domain of secrets and *R* is a randomness set, while $S_{1},\ldots ,S_{n}$ are finite share domains such that the following holds:Correctness: For all A⊆[n] with f(A)=1, referred to as qualified sets, and all $s_{1}\neq s_{2}\in S$, we have:
$$support\left(Sh_{r\leftarrow R}\left(s_{1},r\right)\left[A\right]\right)\cup support\left(Sh_{r\leftarrow R}\left(s_{2},r\right)\left[A\right]\right)\;=\;\phi$$We use sets and their characteristic vectors interchangeably.Privacy: For all A⊆[n] with f(A)=0 and all $s_{1}\neq s_{2}\in S$, the following distributions are equal.
$$Sh_{r\leftarrow R}\left(s_{1},r\right)\left[A\right])=\left(Sh_{r\leftarrow R}\left(s_{2},r\right)\left[A\right]\right)$$

Access structures where the set of qualified subsets is exactly those of size some *t* or more, are called threshold access structures.

We say that a secret sharing scheme is linear over some ring *G*, generalizing the standard notion of linearity over a field, if $S=G,R=G^{l}$ for some l∈N for some finite Abelian ring *G*. Each share consists of one or more linear functions for the form $l\left(s,r_{1},\ldots ,r_{m}\right)=\alpha_{0}\cdot s+\sum_{i}\alpha_{i}\cdot r_{i}$ (Observe that unlike linear schemes over a field, such a scheme does not always have perfect correctness for all secrets s∈R, but this is not required in our context. We will require perfect privacy though.) A useful property of such schemes is that they allow evaluating locally linear functions of the shares. That is, for a pair of sharings $s^{1}=\left(s_{1}^{1},\ldots ,s_{n}^{1}\right)$$s^{2}=\left(s_{1}^{2},\ldots ,s_{n}^{2}\right)$ of some $s_{1},s_{2}$, respectively, $s^{1}+s^{2}$ (performing addition coordinate-wise on each group element in the share vector) is a sharing of $s_{1}+s_{2}$. Similarly, for a∈G, $a\cdot s^{1}$ is sharing of $a\cdot s_{1}$.

See [14] for a survey on secret sharing.

Let us recall the well-known schemes (2,3)-CNF and 3-additive for completeness. The schemes are more general, (2,3)-CNF is a special case of (t,n)-CNF and 3-additive is a special case of (t,n)-DNF with t=n=3. Here we explicitly recall the definitions only of the special cases we need.

A 3-additive secret sharing scheme over a ring *G* is a randomized mapping $f_{ADD}:S\times R\to S_{1}\times S_{2}\times S_{3}$, where $S=S_{i}=G,R=G^{2}$, such that $f_{ADD}\left(s;a_{1},a_{2}\right)=\left(a_{1},a_{2},s-a_{1}-a_{2}\right)$. It is not hard to see that $f_{ADD}$ indeed implements an access structure where only the set {1,2,3} is qualified.

A (2,3)-CNF over a ring *G* is defined by a randomized mapping $f_{CNF}:S\times R\to G^{3}$ for S=G, $S_{i}=R=G\times G$. It is defined as follows $f_{CNF}\left(s;a_{1},a_{2}\right):$Calculate a three-additive sharing $\left(a,b,c\right)=f_{ADD}\left(s;a_{1},a_{2}\right)$.Output shares $\left(s_{1},s_{2},s_{3}\right)$ where $s_{i}$ equals the two elements from the tuple in one, which are not at its index. For example, $s_{3}=\left(a,b\right)$.

It is not hard to see that (2,3)-CNF is indeed a 2-threshold scheme. It is also not hard to see that the above schemes are linear over their respective rings.

A (2,n)-modified Shamir scheme over a ring *G* is defined by a randomized mapping $f_{Sh}\left(s;z\right):G\times G\to G$ outputs shares $\left(s_{1},\ldots ,s_{n}\right)$
$s_{i}\;=\;z+sx_{i}$, where $x_{i}\in G$ is a distinct constant ‘evaluation point’.

It is not hard to see that (2,3)-CNF is indeed a two-threshold scheme. It is also not hard to see that the above schemes are linear over their respective rings.

### 3.2. Share Conversion

We recall the definition of (generalized) share conversion schemes as considered in our paper. Our definition is exactly the definition in [4], in turn adopted from previous work.

**Definition** **1** ([4])**.**
*Let $L_{1}$ and $L_{2}$ be two n-party secret-sharing schemes over the domains of secrets $K_{1}$ and $K_{2}$, respectively, and let $C\subseteq K_{1}\times K_{2}$ be a relation such that, for every $a\in A_{1}$, there exists at least one $b\in K_{2}$ such that (a,b)∈C. A share conversion scheme $convert(s_{1},\ldots ,s_{n})$ from $L_{1}$ to $L_{2}$ with respect to relation C is specified by (deterministic) local conversion functions $g_{1},\cdots ,g_{n}$ such that: If $\left(s_{1},\cdots s_{n}\right)$ is a valid sharing for some secret s in $L_{1}$, then $g_{1}\left(s_{1}\right),\cdots g_{n}\left(s_{n}\right)$ is a valid sharing for some secret $s^{\prime }$ in $L_{2}$ such that $(s,s^{\prime })\in C$.*
*(In [4], they referred to such share conversion schemes as “local” share conversion. As this is the only type we consider here, we will refer to it as simply “share conversion”.)*


For a pair of Abelian groups $G_{1},G_{2}$ (when $G_{1},G_{2}$ are rings, we consider $G_{1},G_{2}$ as groups with respect to the $‘‘{+}^{\prime \prime }$ operation of the rings), we define the relation $C_{S}$ as in [4] ($G_{1},G_{2}$ will be clear from the context, so we exclude the groups from the definition of $C_{S}$ to simplify notation).

**Definition** **2** (The relation *C_S_* [4])**.**
*Let $G_{1}$ and $G_{2}$ be finite Abelian groups, and let $S\subseteq G_{2}\backslash \left\{0\right\}$ (when $G_{1},G_{2}$ are rings, we will by default refer to the additive groups of the rings in this context). The relation $C_{S}$ converts $s\;=\;0\in G_{1}$ to any nonzero $s\in G_{2}$ and every s∈S to s=0. There is no requirement when s∉S∪0. Formally,*
$$C_{S}\;=\;\left\{\left(s,0\right)|s\in S\right\}\cup \{\left(0,s^{\prime }\right):s^{\prime }\in G_{2}\backslash \left\{0\}\right\}\cup \left\{\left(s,s^{\prime }\right)|s\notin S\cup \left\{0\right\},s^{\prime }\in G_{2}\right\}$$

Given $m=p_{1}\cdot p_{2}$, where $p_{1}\neq p_{2}$ are primes and *p* is a prime, we consider pairs of rings $G_{1}=Z_{m},G_{2}=Z_{p}^{\beta }$. We denote $S_{m}=\left\{x\in G_{1}|\forall i\in \left[2\right],x\;\mathrm{mod}\;p_{i}\in \left\{0,1\right\}\right\}\backslash \left\{0\right\}$.

### 3.3. Our Starting Point: The Modeling of [4]

In this work, we study the existence of share conversions for three parties with respect to the canonical relation $C_{G_{1},G_{2}}$ as above, from (2,3)-CNF to three-additive for various parameters $p_{1},p_{2},p$.

Our starting point is the characterization from [4] of triples $\left(p_{1},p_{2},p\right)$ for which a share conversion with respect to $\left(G_{1},G_{2}\right)$ as above exists via a linear-algebraic constraint.

In some more detail, consider $\left(p_{1},p_{2},p\right)$ as above. A share conversion from (2,3)-CNF to three-additive exists for $S_{m}$ iff a certain condition is satisfied by the following matrix $M_{\equiv ,≢}$ over $Z_{p}$.

In the matrix $M_{\equiv ,≢}$, the rows are indexed by triples $\left(a,b,c\right)\in Z_{m}^{3}$, corresponding to (2,3)-CNF sharings of some s∈S∪{0}. Namely, (a,b,c) are the (additive) shares generated by $f_{CNF}$ in Step 1. The rows corresponding to s≠0 form $M_{\equiv }$. The rows corresponding to s=0 form $M_{≢}$. The columns of $M_{\equiv ,≢}$ are indexed by values in $\left[3\right]\times Z_{m}\times Z_{m}$. Intuitively, an index (i,x,y) of a column corresponds to share $s_{i}$ of the (2,3)-CNF scheme being equal to (x,y). Row (a,b,c) has ones at three locations: (3,a,b),(2,a,c),(1,b,c), and zeros elsewhere. That is, there are zeros at columns corresponding to the shares $s_{i}$ output by $f_{CNF}$ corresponding to (a,b,c) produced in Step 1 of $f_{CNF}$’s execution.

We are searching for a vector *u* that by multiplying it with each row of the “equality rows” of the matrix, it will be equal zero, and by multiplying it with one of the inequality rows, it will not be equal to zero.

The solution vector *u* (and thus, the columns of the matrix) is indexed by $\left[3\right]\times Z_{m}\times Z_{m}$. The index of an entry (i,x,y) corresponds to a CNF-share $s_{i}$ that equals (x,y), and the value $u_{\left(i,x,y\right)}$ at this index is the value in $F_{b^{\beta }}$ to which share $s_{i}=\left(x,y\right)$ is converted.

Indeed, it is not hard to see that a share conversion scheme exists iff a solution $x\in F_{p^{\beta }}$ to the system:$$\left\{\begin{array}{c} M_{\equiv }x=0\\ M_{≢}x\neq 0 \end{array}\right.$$
exists.

Some basic linear-algebraic observations imply that the above is equivalent to the fact that the rows of $M_{\equiv }$ do not span some row of $M_{\equiv }$ over $F_{p}$ (this simplification matters, as it does not require us to know β in advance, if it exists).

Furthermore, the work in [4] provided a quantitative lower bound on β, depending on the degree difference between $M_{\equiv }$ and $M_{≢}$ (the latter is not significant towards our goal of just understanding feasibility). Their characterization is summarized in the following theorem.

**Theorem** **3** (Theorem 4.5 [4])**.**
*Let $\beta \;=\;rank_{F_{p}}\left(M_{\equiv ,≢}\right))-rank_{F_{p}}\left(M_{\equiv }\right))>0$. Then, we have:*
If β=0, then there is no conversion from (2,3)-CNF sharing over $Z_{m}$ to additive sharing over $Z_{p}^{\kappa }$ with respect to $C_{S_{m}}$, for every κ>0.If β>0, then there is a conversion from (2,3)-CNF sharing over $Z_{m}$ to additive sharing over $Z_{p}^{\beta }$ with respect to $C_{S_{m}}$.Furthermore, in this case, every row v of $M_{≢}$ is not spanned by the rows of $M_{\equiv }$.

Moreover, it was proven in [4] that the above is in fact equivalent to having the rows of $M_{\equiv }$ not span any row of $M_{\equiv }$. The latter simplification is a result of certain symmetry existing for the particular relation and secret sharing schemes in question.

**Corollary** **1.**
*For every row v of $M_{≢}$, v is not spanned by the rows of $M_{\equiv }$ iff there exists some β>0 for which a conversion from (2,3)-CNF sharing over $Z_{m}$ to additive sharing over $Z_{p}^{\beta }$ with respect to $C_{S_{m}}$ exists.*


Theorem 3 provides a full characterization via a condition that given $\left(p_{1},p_{2},p\right)$ can be verified in polynomial time in $\left(p_{1},p_{2},\log\left(p\right)\right)$. More precisely, the size of our matrix $M_{\equiv ,≢}$ is $4m^{2}\times 3m^{2}$ (in fact slightly smaller, at $\left(3m^{2}+1\right)\times 3m^{2}$, if working with Corollary 1, but this is insignificant), so verifying the condition amounts to solving a set of linear equations, which naively takes about $O(m^{6})$ time, or slightly better using improved algorithms for matrix multiplication, and the running time cannot be better than $\tilde{\Omega }\left(m^{4}\right)$ using generic matrix multiplication algorithms. Thus, the complexity of verification grows very fast with *m*, becoming essentially infeasible for $p_{1},p_{2}$ circa 50.

In any case, direct verification of the condition on concrete inputs does not answer the following fundamental question.

**Question** **1** (Informal)**.**
*Do there exist infinitely many triples $\left(p_{1},p_{2},p\right)$ for which a share conversion scheme from (2,3)-CNF to three-additive secret sharing (with parameters as discussed above) exists?*


It was conjectured in [4] that all tuples where $p\;=\;p_{1}$ and $p_{1},p_{2},p$ are all odd allow for such a sharing scheme. We answer this question in the affirmative. While it is not clear that our result may be directly useful towards constructing better PIR schemes, our work is a first step towards possibly improving PIR complexity using this direction, in terms of the tools developed. See the discussion in Section 5 for more details.

To resolve this question, we develop an analytic understanding of the condition for the particular type of matrices at hand. Our goal is to simplify the matrices into a more human-understandable form, so we are able to verify directly the linear-algebraic condition for infinitely many parameter triples.

### 3.4. Some Notation

In this paper, we will consider matrices over some field F, typically over a finite field $F=Z_{p}$. For matrices A,B with the same number of columns, (A;B) denotes the matrix comprised by concatenating *B* below *A*. For matrices A,B with the same number of rows, we denote by (A|B) the matrix obtained by concatenating *B* to the right of *A*. For entry i,j of a matrix *A*, we use the standard notation of A[i,j]. More generally, for a matrix $A\in F^{u\times v}$, for subsets R⊆[v] of rows and C⊆[u] of columns, A[R,C] denotes the sub-matrix with rows restricted to *R* and columns restricted to *C* (ordered in the original order of rows and columns in *A*). As special cases, using a single index *i* instead of *R* (*C*) refers to a single row (column). A “·” instead of *R* (*C*) stands for [u] ([v]).

We often consider imposing a block structure upon a matrix *A*. The block structure is specified by a grid defined by a partition of the columns into non-empty sets of consecutive columns $C_{1},\ldots ,C_{t}$ and a partition of the rows into non-empty sets of consecutive rows $R_{1},\ldots ,R_{h}$. The matrix *A* viewed as a block matrix is not a t×h matrix where entry (i,j) is the sub-matrix $A[C_{i},R_{i}]$. We denote the block matrix obtained from *A* by *V* (for instance, a matrix named $A^{\left(3\right)}$ is replaced by $V^{\left(3\right)}$). In a block matrix *V*, we typically index the matrix by subscripts: $V_{i,j}$ denotes (the matrix at) entry i,j of *V*. For instance, $V_{i,j}\left[i,k\right]$ denotes entry [i,k] in (sub)matrix $V_{i,j}$.

Typically, the $C_{i}$’s and the $R_{i}$’s are of the same size, and $C_{0},R_{0}$ start at zero. Sometimes, the sets will not be of the same size (typically, the first or last set will be of a different size than the rest). Furthermore, most generally, $C_{0},R_{0}$ may start elsewhere. Additionally, the indices may be consecutive modulo *u* (*v*), so one of the sets $C_{i}$ ($R_{i}$) may not consist of truly consecutive indices.

Most of the time, index arithmetic will be done modulo the matrices’ number of rows and columns (we will however state this explicitly).

## 4. Our Result

### 4.1. Starting Point and Main Technical Tool

Starting off with Corollary 1, it suffices to prove the following theorem.

We prove the following result.

**Theorem** **4.**
*Assume $m=p_{1}\cdot p_{2}$, $p=p_{1}$, and $p_{1},p_{2}>2$. Then, there exists a row v in $M_{≢}$ such that $Rows(M_{\equiv })$ does not span v.*


As a corollary from Corollary 1 and Theorem 4, we immediately obtain:

**Corollary** **2.**
*Assume $m=p_{1}\cdot p_{2}$, $p=p_{1}$, and $p_{1},p_{2}>2$. Then, there exists some β>0 for which a conversion from (2,3)-CNF sharing over $Z_{m}$ to additive sharing over $Z_{p}^{\beta }$ with respect to $C_{S_{m}}$ exists.*


To prove Theorem 4, we choose any vector *v* in the $M_{\equiv ,≢}$, outside of $M_{\equiv }$, and prove that $M_{\equiv }$ does not span *v*. The particular choice of *v* we will make is a somewhat convenient choice, but any *v* will do, so we will fix it later. To this end, we apply carefully-chosen row operations, but in a specific way, so we (as humans) can understand the matrices that result.

Our main technical tool is the following simple lemma.

**Lemma** **1.**
*Let A denote a matrix in $Z_{p}^{v\times u}$, and let b=A[v,[u]]. Let $I_{1}\subseteq \left[v-1\right],I_{2}\subseteq \left[u\right]$ denote non-empty sets of rows and columns, respectively. $A^{\prime }$ obtained from A by a sequence of row operations on A, so that $A^{\prime }\left[I_{1},I_{2}\right]$ is a basis of $A^{\prime }\left[\left[v\right],I_{2}\right]$, and the rest of the rows in $A^{\prime }\left[I_{1},I_{2}\right]$ are zero. Then, $Rows(A^{\prime }\left[\left[v\right]\backslash I_{1},\left[u\right]\backslash I_{2}\right])$ span $b^{\prime }\left[\left[u\right]\backslash I_{2}\right]$ iff Rows(A[[v−1],[u]]) span A[v,[u]].*


The proof of the Lemma follows by the fact that any solution to xA[[v−1],[u]]=A[v,[u]] must have zero at indices corresponding to $I_{1}$, to obtain zero at the coordinates corresponding to $v_{2}$ (and the fact that row operations on [A;b] do not change the solvability of a system Ax=b).

We start with $M_{\equiv }^{\prime }=\left[M_{\equiv };v\right]$ and need to resolve the question of whether $Rows(M_{\equiv })$ span *v*. On a high level, we proceed by applying the following lemma several times, thereby reducing the problem to considering a certain submatrix of the original matrix.

### 4.2. A Few Initial Elimination Sequences

#### 4.2.1. Elimination Step 0

We introduce some more notation we will use along the way. We think of the matrix $M_{\equiv ,≢}^{\prime }$ as divided into blocks of 4×3. We denote constants by capital letters (e.g., a secret value $S_{1}$, or index (A,B)) and running indices by small letters, typically a,b,c. Row 1 of the block matrix *V*, $V_{i,\cdot }$ is indexed inside by (a,b,c), going over all (a,b,c) that constitute an additive sharing of $S_{1}={\left(0,1\right)}_{Z_{m}}$. Here, ${\left(0,1\right)}_{Z_{m}}$ denotes a single element of $Z_{m}$, corresponding to its residues modulo $p_{1}$ and $p_{2}$, respectively. We omit the $Z_{m}$ subscript when it is clear from the context whether a pair (x,y) is in $Z_{m}$ or in $Z_{m}$. Similarly, rows $V_{2,\cdot },V_{3,\cdot },V_{4,\cdot }$ correspond to $S_{2}={\left(1,0\right)}_{Z_{m}},S_{3}={\left(1,1\right)}_{Z_{m}},S_{4}={\left(0,0\right)}_{Z_{m}}$, respectively. The rows inside rows 1,2,3 in the block matrix are internally indexed by (a,b) (*c* is determined by $a,b,S_{i}$ as $S_{i}-a-b$), where the (a,b)’s are ordered lexicographically (we stress that the particular order is not important for analyzing the rank of the matrix, but it will play some role in creating a matrix that “looks simple” and is easier to understand). The last block is indexed by some $\left(A,B\right)\in Z_{m}\times Z_{m}$ to be chosen later; as follows from Corollary 1, any (A,B) can be chosen.

Column $V_{\cdot ,1}$ in the block matrix is internally indexed by the CNF-share (a,b) received by $P_{3}$. Column $V_{\cdot ,2}$ is indexed by (a,c) (share received by $P_{2}$), and $V_{\cdot ,3}$ is indexed by (b,c) (share received by $P_{3}$). As in the case of rows, the values $\left(x,y\right)\in Z_{m}$ internally indexing a column of *V* are ordered lexicographically.

Pictorially, the matrix $M_{\equiv }^{\prime }$, has the following general form.



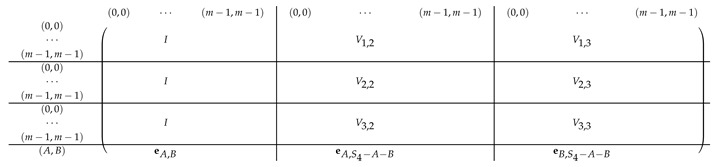



In the above, *I* denotes the identity matrix, and each $V_{i,1}$ indeed equals *I*. The other $V_{i,j}$’s for j∈{2,3} are $m^{2}\times m^{2}$ permutation matrices, and $e_{x,y}\in Z_{p}^{m\times n}$ denotes a row vector with one at index (x,y), and zeroes elsewhere.

For a fixed i∈[3], row (a,b) equals $\left(e_{a,b},e_{a,S_{i}-a-b},e_{b,S_{i}-a-b}\right)$ for Block Columns 1,2,3 respectively.

Next, we apply Lemma 1 to $A=M_{\equiv }^{\prime }$ with $I_{1}=\left[m\right]$, and $I_{2}=\left[m^{2}\right]$ ($I_{2}$ corresponds to the first column in the block). The sequence of operations to b-zero the columns in $A[\left[3m^{2}+1\right],\left[m^{2}\right]]$ is carried out by subtracting from each row $e_{a,b}$ (indexed by (a,b)) in Blocks 2,3,4 the corresponding row $e_{a,b}$ (indexed by (a,b)) in Block 1. The resulting matrix $A^{\prime }$ (of size $\left(2m^{2}+1\right)\times 2m^{2}$) is described in the following.

#### 4.2.2. Elimination Step 1

We view the matrix $A^{\prime }$ as a block matrix with the same subdivision into blocks as in $M_{\equiv }^{\prime }$ (with one less block in both rows and columns, with Row 1 and Column 1 removed). Let $V^{\prime }$ denote the corresponding 3×2 block matrix.

Consider the row indexed by (a,b) in $V_{i,\cdot }^{\prime }$ for i∈[3] (corresponds to Block Rows 2,3,4 respectively in $M_{\equiv }^{\prime }$). We have:

**Observation** **1.**
*The row $V_{i,\cdot }^{\prime }$ indexed by (a,b) - $V_{i,\cdot }^{\prime }\left[\left(a,b\right),\cdot \right]$ equals:*
(1)$$\left(e_{a,S_{i}-a-b}-e_{a,S_{1}-a-b};e_{b,S_{i}-a-b}-e_{b,S_{1}-a-b}\right)$$
*Here, (a,b)=(A,B) for i=3.*


The resulting matrix $A^{\prime }$ is depicted next.



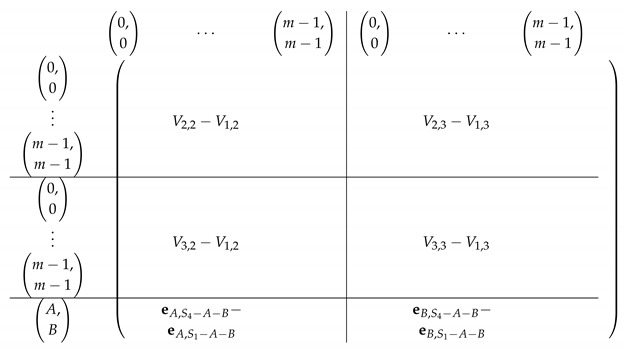



Next, we find a basis for $Rows(A\left[I_{1},\left[m^{2}\right]\right])$ and apply Lemma 1 again, “getting rid” of the first block column in $A^{\prime }$. For i∈{1,2,3,4}, let $\Delta_{i,j}=S_{i}-S_{j}$.

We will need another simple observation.

**Observation** **2.**
*The rows of $V_{i,1}^{\prime }$ for i∈[1,2] are a permutation of the sequence of all $m^{2}$ vectors of the form $\left(x,y\right)-\left(x,y+\Delta_{1,i+1}\right)$. Additionally, $\Delta_{1,2}\;=\;{\left(-1,1\right)}_{Z_{m}}$, and therefore, ${\left(-1,1\right)}_{Z_{m}}$ generates $Z_{m}$.*


We are now ready to demonstrate that the set of rows of $V_{1,1}^{\prime }$, according to the coordinate of one in their row: $B=\{\left(a,b\right)|c=S_{2}-a-b\neq m-1\}$ constitutes the basis of $V_{\cdot ,I_{2}}^{\prime }$ for $I_{2}=\left[m^{2}\right]$, as we seek. Therefore, we will again be able to apply Lemma 1. The other rows are spanned by this set, in one of two ways. We classify them into two types according to these ways. We refer to them as Type 1 and Type 2.

Type 1 constitutes of rows in $V_{1,1}^{\prime }$ indexed by (a,b) with $c=S_{2}-a-b=m-1$, for which the location of the ‘1’ in $V_{1,1}\left[\left(a,b\right),\cdot \right]$ is (a,c) for *c* as above. We observe that every such row is spanned by $V_{1,1}^{\prime }\left[B,\cdot \right]$, as for every $a\in Z_{m}$, we have:(2)$$\begin{array}{c} \sum_{b\in Z_{m}}V_{1,1}^{\prime }\left[\left(a,b\right),\cdot \right]= \end{array}$$
(3)$$\begin{array}{c} \sum_{c\in Z_{m}}e_{a,c}-e_{a,c+\Delta_{1,2}}= \end{array}$$
(4)$$\begin{array}{c} \sum_{c\in Z_{m}}e_{a,c}-\sum_{c\in Z_{m}}e_{a,c+\Delta_{1,2}}= \end{array}$$
(5)$$\begin{array}{c} 1-1=0 \end{array}$$

Here, the transition from Equation (Equation 2) to Equation (Equation 3) is by Observation 1 and the observation that for a fixed *a*, ${\left(S_{2}-a-b\right)}_{b\in Z_{m}}$ is a permutation over $Z_{m}$, so the change of coordinates we perform is valid. We b-zero rows (a,b) of $V_{1,1}^{\prime }$ of Type 1 by adding all other rows of the form $\left(a,b^{\prime }\right)$ to each of them.

Type 2 includes all rows in $V_{i,1}^{\prime }$ for i∈{2,3} are of this type. Consider a row indexed by (a,b) in $V_{i,1}^{\prime }$ (in particular, the row in block $V_{3,\cdot }^{\prime }$ is of that type). It is spanned by rows in B as follows:(6)$$\begin{array}{c} \sum_{j=0}^{\frac{\Delta_{1,i+1}}{\Delta_{1,2}}-1}V_{1,1}^{\prime }\left[\left(a,b+\Delta_{2,i+1}-j\cdot \Delta_{1,2}\right),\cdot \right]= \end{array}$$
(7)$$\begin{array}{c} \sum_{j=0}^{\frac{\Delta_{1,i+1}}{\Delta_{1,2}}-1}\left(e_{a,S_{i+1}-a-b+j\cdot \Delta_{1,2}}-e_{a,S_{i+1}-a-b+\left(j+1\right)\cdot \Delta_{1,2}}\right)= \end{array}$$
(8)$$\begin{array}{c} e_{a,S_{i+1}-a-b}-e_{a,S_{i+1}-a-b+\Delta_{1,i+1}}=V_{i,1}^{\prime }\left[\left(a,b\right),\cdot \right] \end{array}$$

Here, the transition to Equation (Equation 8) is by observing that a telescopic sum is formed, where all but the first and last $e_{x,y}$’s in the sum cancel out. Note that the number in the superscript of the sum indeed exists, as $\Delta_{1,2}$ generates $Z_{m}$, and so, in particular, $\Delta_{1,i+1}=k\cdot \Delta_{1,2}$ for some integer k>0.

Rearranging the equality above, we get:(9)$$\begin{array}{c} V_{i,1}^{\prime }\left[\left(a,b\right),\cdot \right]-\sum_{j=0}^{\frac{\Delta_{1,i+1}}{\Delta_{1,2}}-1}V_{1,1}^{\prime }\left[\left(a,b+\Delta_{2,i+1}-j\cdot \Delta_{1,2}\right),\cdot \right]=0 \end{array}$$

That is, we have identified the linear combinations of rows in B that add up to $V_{i,1}^{\prime }$.

### 4.3. Elimination Step 2

After applying Lemma 1 to each row to Type 1 and Type 2 and making it zero, we turn to considering the remaining matrix $A^{\prime \prime }\in Z_{p}^{\left(m+m^{2}+1\right)\times m^{2}}$. It is a block matrix of size 3×1, with blocks corresponding to $V_{1,2}^{\prime },V_{1,3}^{\prime },V_{1,4}^{\prime }$ in $A^{\prime }$, with $m,m^{2}$, and one row, respectively. Let us calculate the form of rows of both types, when restricted to $V^{\prime \prime }$ (following elimination step 1 from Section 4.2.2).

Type 1 in $V^{\prime \prime }$: From Equation (Equation 2), we conclude that for a fixed *a*, for the (single) *b* such that (a,b)∉B, we have:(10)$$\begin{array}{c} V_{1,1}^{\prime \prime }\left[\left(a,b\right),\cdot \right]= \end{array}$$
(11)$$\begin{array}{c} \sum_{b\in Z_{m}}V_{1,2}^{\prime }\left[\left(a,b\right),\cdot \right]= \end{array}$$
(12)$$\begin{array}{c} \sum_{b\in Z_{m}}\left(e_{b,S_{2}-a-b}-e_{b,S_{1}-a-b}\right) \end{array}$$

Type 2 in $V^{\prime \prime }$: This type consists of row blocks i∈{2,3} in $V^{\prime \prime }$. By construction, we have:(13)$$\begin{array}{c} V_{i,1}^{\prime \prime }\left[\left(a,b\right),\cdot \right]=V_{i+1,2}^{\prime }\left[\left(a,b\right),\cdot \right]-\sum_{j=0}^{\frac{\Delta_{1,i+1}}{\Delta_{1,2}}-1}V_{1,2}^{\prime }\left[\left(a,b+\Delta_{2,i+1}-j\cdot \Delta_{1,2}\right),\cdot \right] \end{array}$$

Fix some $\left(a,b\right)\in Z_{m}\times Z_{m}$. Now, from Observation 1 and Equation (Equation 9), it follows that for all $\left(a,b\right)\in Z_{m}\times Z_{m}$, we have: (14)$$\begin{array}{c} V_{i,1}^{\prime \prime }\left[\left(a,b\right),\cdot \right]= \end{array}$$
(15)$$\begin{array}{c} V_{i,2}^{\prime }\left[\left(a,b\right),\cdot \right]-\sum_{j=0}^{\frac{\Delta_{1,i+1}}{\Delta_{1,2}-1}}V_{1,2}^{\prime }\left[\left(a,b+\Delta_{2,i+1}-j\cdot \Delta_{1,2}\right),\cdot \right]= \end{array}$$
(16)$$\begin{array}{c} e_{b,S_{i+1}-a-b}-e_{b,S_{1}-a-b}\\ -\sum_{j=0}^{\frac{\Delta_{1,i+1}}{\Delta_{1,2}}-1}\begin{array}{c} (\;e_{b+\Delta_{2,i+1}-j\cdot \Delta_{1,2},S_{i+1}-a-b-\Delta_{2,i+1}+j\cdot \Delta_{1,2}}\\ \;-e_{b+\Delta_{2,i+1}-j\cdot \Delta_{1,2},S_{1}-a-b-\Delta_{2,i+1}+\left(j+1\right)\cdot \Delta_{1,2}}) \end{array} \end{array}$$

### 4.4. Permuting the Columns

Next, we permute the columns of $A^{\prime \prime }$, to gain insight into its form. Note that this does not affect the question of whether $A^{\prime \prime }\left[m+m^{2}+1\right]$ (last line) is spanned by the rest of the rows in $A^{\prime \prime }$, $Rows(A^{\prime \prime }\left[\left[m+m^{2}\right],\left[m^{2}\right]\right])$. We refer to the new matrix as $A^{\left(3\right)}$ and the resulting block matrix as $V^{\left(3\right)}$.

The permutation is as follows: the content of a column indexed by $\left(b,c\right)\in Z_{m}\times Z_{m}$ moves to $\left(b,b+c\right)/\Delta_{1,2}$ in the new matrix. We therefore obtain the following matrix *M* for Types 1 and 2, respectively (from Equations (Equation 12) and (Equation 16)).

#### 4.4.1. Type 1

(17)$$\begin{array}{c} V_{1,1}^{\left(3\right)}\left[\left(a,b\right),\cdot \right]=\sum_{b\in Z_{m}}e_{\left(b,S_{2}-a\right)/\Delta_{1,2}}-e_{\left(b,S_{1}-a\right)/\Delta_{1,2}}= \end{array}$$

(18)$$\begin{array}{c} \sum_{b,c\in Z_{m}}e_{\left(b,c\right)/\Delta_{1,2}}-e_{\left(b,c+\Delta_{1,2}\right)/\Delta_{1,2}}= \end{array}$$

(19)$$\begin{array}{c} \sum_{b,c\in Z_{m}}e_{b,c}-e_{b,c+1} \end{array}$$

Here, the last transition is due to the fact that dividing by $\Delta_{1,2}$ is a permutation over $Z_{m}$ (as $\Delta_{1,2}$ is invertible).

#### 4.4.2. Type 2

For i∈{2,3}, we have:(20)$$\begin{array}{c} V_{1,i}^{\left(3\right)}\left[\left(a,b\right),\cdot \right]= \end{array}$$
(21)$$\begin{array}{c} e_{\left(b,S_{i+1}-a\right)/\Delta_{1,2}}-e_{\left(b,S_{1}-a\right)/\Delta_{1,2}}\\ -\sum_{j=0}^{\frac{\Delta_{1,i+1}}{\Delta_{1,2}}-1}\left(e_{\left(b+\Delta_{2,i+1}-j\cdot \Delta_{1,2},S_{i+1}-a\right)/\Delta_{1,2}}-e_{\left(b+\Delta_{2,i+1}-j\cdot \Delta_{1,2},S_{1}-a+\Delta_{1,2}\right)/\Delta_{1,2}}\right)= \end{array}$$
(22)$$\begin{array}{c} e_{b^{\prime },c^{\prime }}-e_{b^{\prime },c^{\prime }+\Delta_{1,i+1}/\Delta_{1,2}}-\sum_{j=0}^{\frac{\Delta_{1,i+1}}{\Delta_{1,2}}-1}(e_{\left(b^{\prime }-j+\Delta_{2,i+1}/\Delta_{1,2},c^{\prime }\right)}-e_{\left(b^{\prime }-j+\Delta_{2,i+1}/\Delta_{1,2},c^{\prime }+1\right)}= \end{array}$$
(23)$$\begin{array}{c} e_{\left(b^{\prime },c^{\prime }\right)}-e_{\left(b^{\prime },c^{\prime }+\Delta_{1,i+1}/\Delta_{1,2}\right)}-\sum_{j=0}^{\frac{\Delta_{1,i+1}}{\Delta_{1,2}}-1}e_{b^{\prime }+j,c^{\prime }}-e_{b^{\prime }+j,c^{\prime }+1} \end{array}$$

In Equation (Equation 21), we simply rename $b^{\prime }=b/\Delta_{1,2},c^{\prime }=\left(S_{i+1}-a\right)/\Delta_{1,2}$. In Equation (Equation 22), we observe that for $j=\frac{\Delta_{1,i+1}}{\Delta_{1,2}}$, we have:$$\begin{array}{c} b^{\prime }-j+\Delta_{2,i+1}/\Delta_{1,2}=\\ b^{\prime }+1+\frac{\Delta_{2,i+1}-\Delta_{1,i+1}}{\Delta_{1,2}}=\\ b^{\prime }+1+\frac{\Delta_{2,1}}{\Delta_{1,2}}=b^{\prime }+1-1=b^{\prime } \end{array}$$

Thus, making a change of coordinates and letting the index *j* in the ∑ in Equation (Equation 23) run from $j=\frac{\Delta_{1,i+1}}{\Delta_{1,2}}-1$ down to zero yield the expression in Line Equation 23.

#### 4.4.3. Summary

We have obtained a matrix with rows of the following form.
Type 1 (contributed by Row Block 1 in $V^{\left(3\right)}$) yields all vectors of the form:
$$\sum_{b\in Z_{m}}e_{b,c}-e_{b,c+1}$$
for $c\in Z_{m}$.Type 2 (contributed by Row Block 2 in $V^{\left(3\right)}$) yields all vectors of the form:
(24)$$\begin{array}{c} e_{\left(b,c\right)}-e_{\left(b,c+\Delta_{1,3}/\Delta_{1,2}\right)}-\sum_{j=0}^{\frac{\Delta_{1,3}}{\Delta_{1,2}}-1}e_{b+j,c}-e_{b+j,c+1}=\\ e_{\left(b,c\right)}-e_{\left(b,c+{\left(1,0\right)}_{Z_{m}}\right)}-\sum_{j=0}^{{\left(0,-1\right)}_{Z_{m}}}e_{b+j,c}-e_{b+j,c+1}= \end{array}$$For all pairs $b,c\in Z_{m}$.The last line of $V^{\left(3\right)}$ is subsequently referred to as Type 3. Similarly to Type 2, it follows from Equation (Equation 23) that the line in the last block (i=3) has the form:
(25)$$\begin{array}{c} e_{\left(B,C\right)}-e_{\left(B,C+{\left(0,1\right)}_{Z_{m}}\right)}-\sum_{j=0}^{{\left(-1,0\right)}_{Z_{m}}}e_{B+j,C}-e_{B+j,C+1} \end{array}$$Here, $B,C\in Z_{m}$ is some pair of constants (corresponding to A,B to be chosen above) to be fixed later.

In the above, ${\left(-1,0\right)}_{Z_{m}}={\left(0,1\right)}_{Z_{m}}-1$ in the sum limit corresponds to “lifting” the element of $Z_{m}$ into Z and viewing as an integer in {0,…,m−1}. Note that no “wrap around” occurs as (0,−1)≠0 in $Z_{m}$, so we get the correct number in the sum limit.

### 4.5. A Change of Basis

Occasionally, we will refer to rows $A^{\left(3\right)}$ as matrices, with rows indexed by *b* and columns by *c*.

As we have witnessed, so far, we have performed applications of Lemma 1 on the original matrix *A*, which was a simple block matrix of size 4×3 blocks, and the upper three row blocks had three cells of size m${}^{2}\;\times$ m${}^{2}$ each and another “block” row with three cells of size 1 × m${}^{2}$ each cell. By performing two applications of the lemma, we obtained a much smaller matrix $A^{\left(3\right)}$. Namely, we obtained a block matrix with 3 × 1 blocks, with cells of size m × m${}^{2}$, m${}^{2}\;\times$ m${}^{2}$, and 1 × m${}^{2}$, respectively. The price we have paid for the reduction in size is that the structure of the matrix has became more complex. In particular, it is no longer clear how to identify additional row sets $I_{1}$ to continue applying the lemma conveniently.

To create a matrix of more manageable form, we perform a change of basis. We suggest the following basis for the row space of $A^{\left(3\right)}$. $B=B_{1}\cup B_{2}$. Here:$$B_{1}=\left\{e_{b,i}-e_{b,i+1}|b\in Z_{m},i\in \left\{0,\ldots ,p_{2}-2\right\},\right\}$$
and:$$B_{2}=\left\{-e_{b,i+j\cdot {\left(1,0\right)}_{Z_{m}}}+e_{b,i+\left(j+1\right)\cdot {\left(1,0\right)}_{Z_{m}}}|b\in Z_{m},i\in Z_{p_{2}},j\in Z_{p_{1}}\backslash \left\{p_{1}-1\right\}\right\}$$

In all indices here and elsewhere, the arithmetic is over $Z_{m}$.

Indeed $B_{1}\cup B_{2}$ is a basis of $Rows(A^{\left(3\right)})$. To prove this, we first note that it is an independent set. Roughly, this follows by separately considering the vectors in *B* with nonzero values in each row (b,·) separately and the fact that $p_{2}$ divides *m*.

Next, we show that the rows of $A^{\left(3\right)}$ are indeed spanned by the above set of vectors. Furthermore, the matrix re-written in this basis will have a nice block structure that we will be able to exploit for the purpose of using Lemma 1.

We denote by $T_{b,i}=e_{b,i}-e_{b,i+1}$ a vector in $B_{1}$ and by $R_{b,i+j{\left(1,0\right)}_{Z_{m}}}=-e_{b,i+j\cdot {\left(1,0\right)}_{Z_{m}}}+e_{b,i+\left(j+1\right)\cdot {\left(1,0\right)}_{Z_{m}}}$ a vector in $B_{2}$.

To rewrite our matrix $A^{\left(3\right)}$, we will specify an ordering of the vectors in *B*, from left to right
The columns in $B_{1}$ come first, in increasing order of *c*, starting from zero.The columns of the form $R_{b,i+j(1,0)}$ in $B_{2}$ are ordered on several levels:
(a)On the highest level, order $R_{b,i+j\cdot (1,0)}$’s according to the index *i* above, starting from zero up to $p_{2}-1$. There are $p_{2}$ blocks of this form on the highest level.(b)For a fixed *i*, order $R_{b,i+j\cdot (1,0)}$’s according to increasing order of *j* starting from zero, up to $p_{1}-2$.(c)For fixed i,j, order in increasing order of $b^{\prime }s$, starting from zero up to m−1.

We order the rows of the matrix as follows:The rows of Type 2 appear first, then Type 1, then Type 3 (the distinguished row to span via the others).Within Type 1, we denote $r_{c}^{1}=\sum_{b\in Z_{m}}e_{b,c}-e_{b,c+1}$.Within Type 2, denote by $r_{b,c}^{2}$ the vector $e_{\left(b,c\right)}-e_{\left(b,c+{\left(1,0\right)}_{Z_{m}}\right)}-\sum_{j=0}^{{\left(0,-1\right)}_{Z_{m}}}e_{b+j,c}-e_{b+j,c+1}$. Consider one such vector, $r_{b,c}^{2}$, We order these vectors according to *i*, then *j*, then *b*, where c=i+j·(1,0). Similarly, for Type 3, denote $r_{B,C}^{3}=e_{\left(B,C\right)}-e_{\left(B,C+{\left(0,1\right)}_{Z_{m}}\right)}-\sum_{j=0}^{{\left(-1,0\right)}_{Z_{m}}}e_{B+j,B}-e_{B+j,C+1}$.

Let us now study the form of the resulting matrix, divided into blocks, as follows from the representation of the various vectors in $Rows(A^{\left(3\right)})$ in basis *B*.

Let us represent the matrix as a block matrix, then we further break down each block into lower level blocks as follows. Let us denote the new matrix by $A^{\left(4\right)}$.

The Type 2 set of rows in $A^{\left(4\right)}$ has structure as depicted in the following matrix. $A^{\left(4\right)}=\left(A^{(4),2};A^{(4),1},A^{(4),3}\right)$. Let us describe each of the matrices $A^{(2),i}$ below.

#### 4.5.1. The Matrix $A^{(4),2}$


$$A^{(4),2}=\left(A^{(4),L,2}|A^{(4),R,2}\right)$$


Here, the “right side” $A^{R,2}$ is a $p_{2}\times p_{2}$ block matrix. Its contents are as follows.



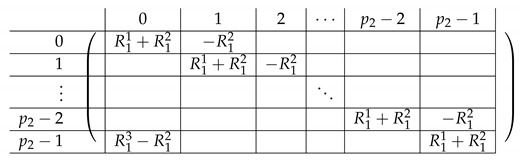



The left-side matrix $A^{(4),L,2}$ is a block matrix of size $p_{2}\times 1$ (where indeed the number of rows in each of its $p_{2}$ blocks is consistent with that of $A^{(4),R,2}$). It has the following structure.



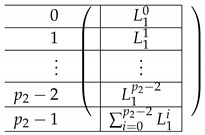



We refer to this partition into $p_{2}\times \left(p_{2}+1\right)$ blocks of $A^{(4),2}$ as the “Level-1” partition of $A^{(4),2}$.

We continue next with describing the “Level-0” detail of $R_{2}^{1},R_{2}^{2},R_{2}^{3},L_{2}^{i}$ of $A^{(4),R,2},A^{\left(4\right),L_{2}}$. The matrix $L_{1}^{i}$ for $i\in \{0,\ldots ,p_{2}-2\}$ is a $p_{1}\times \left(p_{2}-1\right)$ block matrix of the following form:

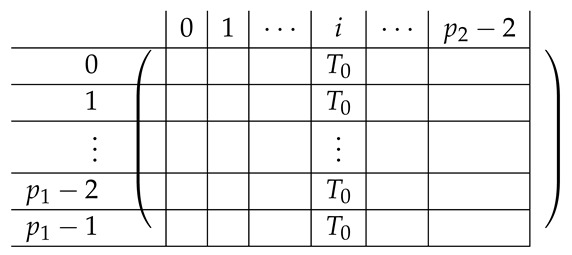

for a matrix $T_{0}$ to be specified later. Note that by the structure of $A^{(4),L,2}$, the last matrix $L_{1}^{p_{2}-1}$ is a block matrix of size $p_{1}\times \left(p_{2}-1\right)$, where each block equals $-T_{0}$.

The matrix $R_{1}^{1}$ is a $p_{1}\times \left(p_{1}-1\right)$ matrix of the following form.



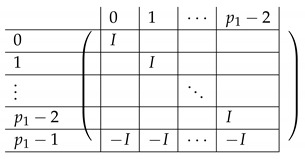



Here, *I* is the m×m identity matrix. The matrix $R_{1}^{2}$ is a $p_{1}\times \left(p_{1}-1\right)$ matrix of the following form.



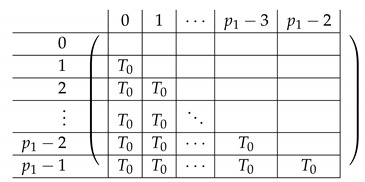



Here, $T_{0}$ is a m×m matrix, as above. Finally, the matrix $R_{2}^{3}$ is a $p_{1}\times \left(p_{1}-1\right)$ block matrix of the following form.



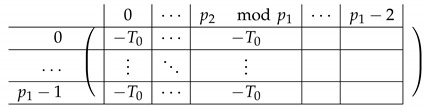



It remains to specify $T_{0}$. It is a m×m circulant matrix of the following form.



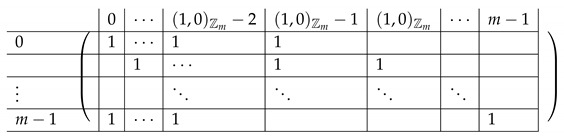



#### 4.5.2. The Matrix $A^{(4),3}$

We choose (B,C)=(0,0). Then, the line is of the form:$$\sum_{b=1}^{{\left(0,1\right)}_{Z_{m}}-1}T_{b,0}+\sum_{j=0}^{p_{1}-2}R_{0,1+j\cdot {\left(1,0\right)}_{Z_{m}}}$$

#### 4.5.3. The Matrix $A^{(4),1}$

The matrix $A^{(4),1}$ is of the form $A^{(4),1}=\left(A^{(4),L,1};A^{(4),R,1}\right)$. To describe the left and right parts, we apply a certain transformation to $A^{(4),L,2}$ and $A^{(4),R,2}$, respectively. First, view each as a block matrix comprised of blocks of size m×m ($A^{(4),L,2}$ has $m\times (p_{2}-1)$ blocks, and $A^{(4),R,2}$ has $m\times p_{2}\left(p_{1}-1\right)$). Now, $A^{(4),L,1}$ is obtained from $A^{(4),L,2}$ by applying a linear mapping satisfying $l\left(T_{0}\right)=\underset{m\;\mathrm{times}}{\underset{︸}{\left(1,\ldots ,1\right)}}$ to each block *X*, replacing *X* by m(X) (it is not important how exactly we define it on other inputs). The matrix $A^{(4),L,1}$ is obtained from $A^{(4),L,1}$ by replacing each block by a linear transformation that maps *I* to the zero vector and $T_{0}$ to $\underset{m\;\mathrm{times}}{\underset{︸}{\left(1,\ldots ,1\right)}}$. That is, the resulting $A^{(4),L,1}$ equals:

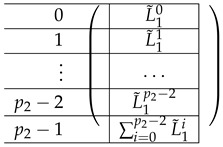
 where ${\tilde{L}}_{1}^{i}$ equals:

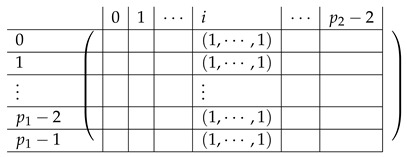


The resulting matrix $A^{(4),R,1}$ equals:

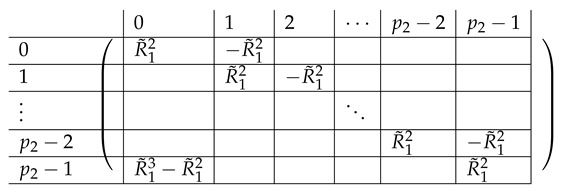
 where the resulting ${\tilde{R}}_{1}^{2}$ is of the form:

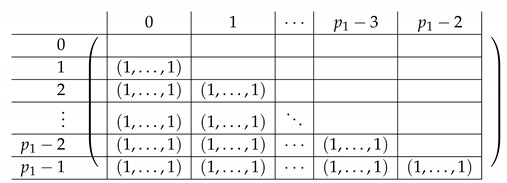


### 4.6. Another Elimination Sequence

From now on, assume that $p=p_{1}$ and that $p_{2}>2$. We leave the full analysis of other cases for future work. We are now able to apply Lemma 1. We perform this step for $I_{2}$ corresponding to the *L*-part blocks of $A^{\left(4\right)}$ and proceed in several steps. We perform the row operations starting at a grosser resolution and then proceed to finer resolution.

#### 4.6.1. Step 1: Working at the Resolution of Level-1 Blocks

View $A^{(4),2}$ as a block matrix of Level-1 as described above. Let $V^{(4),2}$ denote the corresponding block matrix. Replace the last row of $V^{(4),2}$, $V^{(4),2}\left[p_{1}-1,\cdot \right]$ by $\sum_{i=0}^{p_{2}-1}V^{(4),2}\left[i,\cdot \right]$. We thus obtain a new matrix $A^{(5),2}$ of the following form. $A^{(5),2}=\left(A^{(5),L,2}|A^{(5),R,2}\right)$ has the same block structure as $A^{(4),2}$ on all levels, so we do not repeat that, but rather only review its content.

The resulting right side $A^{(5),R,2}$ is as follows.



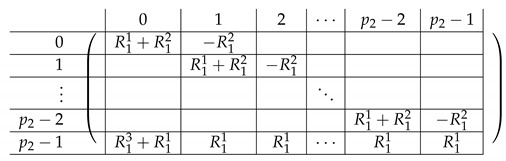



The resulting left-side matrix $A^{(5),L,2}$ is:

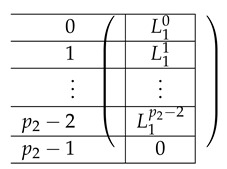


We perform a similar transformation on $A^{(4),1}$, resulting in:$$A^{(5),1}=\left(A^{(5),R,1}|A^{(5),L,1}\right)$$
where $A^{(5),R,1}$ equals:

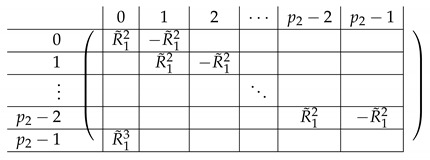
 and $A^{(5),L,1}$ equals



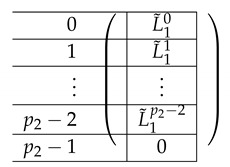



#### 4.6.2. Step 2: Working at the Resolution of Level-0 Blocks

Here, we view the matrix $A^{\left(5\right)}$ as a block matrix over Level-0 blocks. That is, denote by (i,j) the row block corresponding to the $j^{\mathrm{th}}$ Level-0 block inside the $i^{\mathrm{th}}$ Level-1 block of $A^{\left(6\right)}$. We transform $A^{\left(5\right)}$ into a matrix $V^{\left(6\right)}$ as follows.

For each $i\in \left\{0,\ldots ,p_{2}-2\right\},j\in \left\{1,\ldots ,p_{1}\right\}$, replace each row in $V^{(5),2}\left[\left(i,j\right),\cdot \right]$ with $V^{(5),2}-\left[\left(i,0\right),\cdot \right]$. The resulting matrix $A^{(6),2}$ is of the form $A^{(6),2}=A^{((6),L,2}\left|A^{(6),R,2}\right)$.

The right side $A^{(6),R,2}$ is as follows.

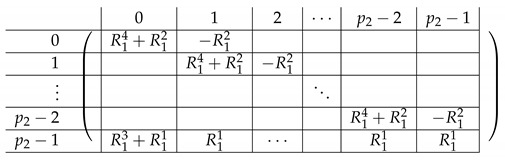
 where $R_{1}^{4}$ equals:



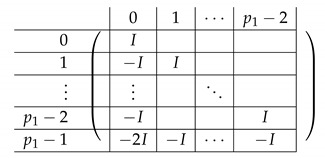



The resulting left-side matrix $A^{(6),L,2}$ is:

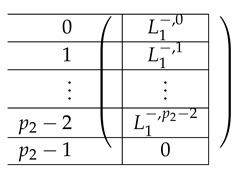
 where $L_{1}^{-,i}$ is of the form:

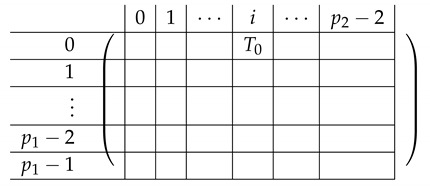


Finally, we apply a similar transformation to $A^{(5),1}$ resulting in $A^{(6),L,1}$ that equals:

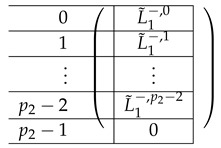
 where ${\tilde{L}}_{1}^{-,i}$ is of the form:

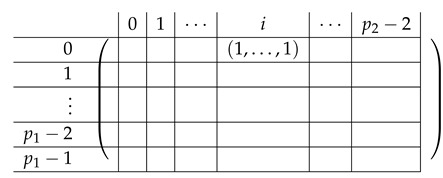


The resulting right-hand side is $A^{(6),R,1}=A^{(5),R,1}$ (there is no change, since the first block-row in $V^{(5),R,1}$ is zero).

#### 4.6.3. Step 3: Working within Level-0 Blocks

Here, we move to working with individual rows and complete the task of leaving a basis of the original $A^{(4),L}$’s rows as the set of non-zero rows of the matrix $A^{(7),L}$ obtained by a series of row operations. To this end, our goal is to understand the set of remaining rows in $A^{(6),L}$. In the $A^{(6),L,2}$ part, each Level-0 block-column (with blocks of size m×m) has only $G=Rows\left(T_{0}\right)\cup \left\{\left(1,\ldots ,1\right)\right\}$ (here, one appears *m* times) as non-zero rows, and in each row, there are non-zero entries in only one of the blocks. Thus, it suffices to find a basis for the set *G* of vectors.

**Lemma** **2.**
*Assume $p=p_{1}$. Then, the index set $I=\{k|0\leq k\leq \left(p_{1}-1\right)p_{2}\}$ satisfies that $Rows(T_{0}\left[I,\left[m\right]\right])$ is a basis for G. In particular, we have $\sum_{j=0}^{p_{1}-1}T_{0}\left[j\cdot p_{2},\left[m\right]\right]=x\cdot \left(1,\ldots ,1\right)$. Here, x is computed as follows: first calculate y as $p_{2}^{-1}$ modulo $p_{1}$ (that is, in $Z_{p_{1}}$). Then, we “lift” y back into Z ($1\leq y\leq p_{1}-1$) and then set x to be y modulo p – that is, x is an element of $Z_{p}$ (note that all non-zero coefficients in the linear combination that results in (1,…,1) indeed belong to I).*


Another observation that will be useful to us identifies the dual of $T_{0}$.

**Lemma** **3.**
*Assume $p=p_{1}$. Then, the set of vectors:*
$$S=\{\sum_{j=0}^{p_{1}-1}e_{j\cdot p_{2}}-e_{i+\left(j+1\right)\cdot p_{2}}|i\in Z_{p_{2}}\backslash \left\{0\right\}\}$$
*is a basis of $Ker(T_{0})$, where Ker(M)={v|v·M} denotes the left kernel of the matrix M.*


The observations are rather simple to prove by basic techniques; see Appendix A. Note that the general theory of cyclotomic matrices is not useful here, as it holds over infinite or large (larger than matrix size) fields, so we proceed by ad-hoc analysis of the (particularly simple) matrices at hand.

We handle the $A^{(7),2}$ part first. We conclude from Lemma 2 that for every block specified by (i,j) where $i\neq p_{2}-1$, in $V^{(6),L,2}\left[\left(i,j\right),\cdot \right]$, the rows indexed by b∈I (as in Lemma 2) span all rows in that block. Furthermore, for the purpose of Lemma 1, we b-zero the rest of the rows, by a sequence of row operations as specified by $Ker(T_{0})$ in Lemma 3, starting from row $\left(p_{1}-1\right)p_{2}+1$ and moving forward up to m−1. That is, for b -zeroing row $\left(p_{1}-1\right)p_{2}+k$ (where k>0) in $V^{(6),L,2}\left[\left(i,j\right),\cdot \right]$ as above, we store the combination:$$\sum_{h=0}^{p_{1}-1}V^{(6),1}\left[\left(i,j,k+h\cdot p_{2}\right),\cdot \right]-V^{(6),1}\left[\left(i,j,k+\left(h+1\right)\cdot p_{2}\right),\cdot \right]$$
in row (i,j,k) of $A^{(7),2}$.

Overall, the resulting $A^{(7),2}$ is as follows:

$A^{(7),R,2}$ is identical to $A^{(6),R,2}$, except for replacing $R_{1}^{4}$ with $R_{1}^{5}$.

That is, in the last block row $R_{1}^{5},$ all cells are $R_{-1}^{2}$, and there are $p_{1}$ such cells.

Here, $R_{-1}^{2}$ is of the form:

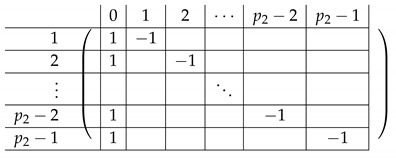

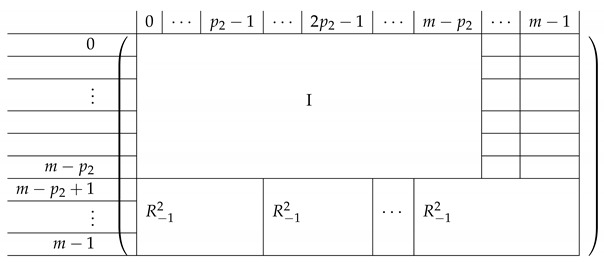


Next, we handle the $A^{(7),1}$ part. Here, we b-zero the remaining rows in $A^{(6),L,1}$ by adding the right combination of rows in $A^{(6),L,2}$. The combination is determined by the “in particular” part of Lemma 3.

The resulting matrix $A^{(7),L,2}$ is identical to $A^{(6),L,2}$, except for $T_{0}$ in each $L_{1}^{-,i}$ being replaced by $T_{0}^{\prime }$. Here, $T_{0}^{\prime }$ has the form:

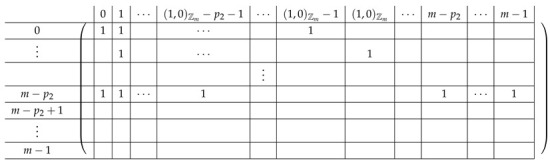


Here, $A^{(7),L,1}$ becomes zero, which was our goal. Note that as opposed to previous transformations, the transformation performed on $A^{(6),L,1}$ does not “mirror” the transformation performed on $A^{(6),L,2}$ and in fact involves rows from both $A^{(6),L,2}$ and $A^{(6),L,1}$. $A^{(7),R,1}$ is identical to $A^{(6),R,1}$, except that in each Level-1 block (i,i) for $i\in \{0,\ldots ,p_{2}-2\}$, the first row of ${\tilde{R}}_{1}^{2}$ (the content of this block) is replaced by:$$-\sum_{i=0}^{p_{1}-1}xe_{i\cdot p_{2}}.$$

It remains to b-zero the *L*-part of $A^{(6),3}$. For simplicity, we focus on $V^{(6),L,3}\left[0,0\right]$ (which is the only non-zero block in $V^{(6),L,2}$) and then use the resulting linear dependence to produce the new row $V^{(6),3}\left[0,\cdot \right]$.
$$V^{(6),L,3}\left[0,0\right]=x\sum_{i=0}^{p_{1}-1}V^{(6),L,2}\left[\left(0,0,i\cdot p_{2}\right),0\right]-V^{\left((6),L,2\right)}\left[\left(0,0,{\left(-1,0\right)}_{Z_{m}}+1\right),0\right]$$

This results in:(26)$$\begin{array}{c} A^{(7),R,3}=-x\sum_{i=0}^{p_{1}-1}e_{\left(0,0,i\cdot p_{2}\right)}+e_{\left(0,0,{\left(-1,0\right)}_{Z_{m}}\right)}+\\ \sum_{b=1}^{{\left(0,1\right)}_{Z_{m}}-1}T_{b,0}\left[R\right]+\sum_{j=0}^{p_{1}-2}R_{0,1+j\cdot {\left(1,0\right)}_{Z_{m}}}\left[R\right] \end{array}$$

#### 4.6.4. A Reduced Matrix We will Analyze Directly

Taking $I_{1}$ to be the set of rows in $A^{\left(7\right)}$ that correspond to non-zero rows in $A^{(7),L,1}$ and $I_{2}$ corresponding to *L*, we obtain the following matrix $A^{\left(8\right)}$. On Level-1, it has a block structure similar to that of $A^{(7),R}$ (where the number of rows changes in some of the matrices). More concretely, $A^{(8),2}$ has the form:

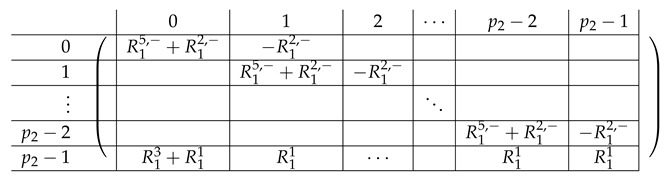


Here, $R_{1}^{5,-}$ is identical to $R_{1}^{5}$ except that the top $m-p_{2}+1$ rows in it are removed. That is, it is identical to $R_{1}^{4}$, except that the (0,0)^th^ Level-0 block in $R_{1}^{4}$ replaces *I* by *C*, which are equal.



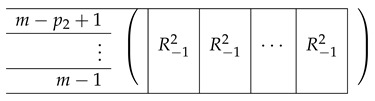



Similarly, $R_{1}^{2,-}$ is obtained from $R_{1}^{2}$ in the same manner. In this case, only zero rows are removed. $A^{(8),1}$ is precisely $A^{(7),R,1}$ (no rows were eliminated from there, as all corresponding rows on the left side became zero). Similarly, $A^{(8),3}=A^{(7),R,3}$.

### 4.7. Completing the Proof - Analysis of $A^{\left(8\right)}$

We are now ready to make our conclusion, assuming $p=p_{1}$ and $p_{1},p_{2}>2$. We stress that further analysis of the matrix is needed for identifying all *p*’s for which a share conversion exists. In fact, some of the detailed calculations of the resulting matrix structure are not needed for our conclusion, and we could instead identify only the properties that we need of various sub-matrices. However, some of the details may be useful for future analysis, so we made all the calculations.

Our last step is to reduce the matrix $A^{\left(8\right)}$ “modulo” the set *G*: for every row *r* in $A^{\left(8\right)}$ and every Level-0 block in this row, we reduce the contents of that row “modulo” $span(Rows\left(T_{0}\right))$. That is, we complement the basis of $Rows(T_{0})$ specified in Lemma 2 into a basis of $Z_{p}^{m}$, where $e_{0}$ is one of the added vectors and define a linear mapping *L* taking elements of $Rows(T_{0}\left[I,\cdot \right])$ to zero and other elements of the basis onto themselves (it is inconsequential what the other base elements are). Indeed, observe that $e_{0}$ is not in $span(Rows\left(T_{0}\right))$, as it is not in $Ker(Ker\left(span\left(Rows\left(T_{0}\right)\right)\right))$, as implied by Lemma 2. To verify this, observe for instance that:$$<\sum_{i=0}^{p_{1}-1}\left(e_{i\cdot p_{2}}-e_{1+i\cdot p_{2}}\right),e_{0}>=1\neq 0.$$

We apply a linear mapping *L* taking $x\in span(Rows\left(T_{0}\right))$ to 0 and other base elements to themselves. Recall that Level-0 blocks indeed have *m* columns each. We make the following observations. We let $A^{\left(9\right)}$ denote the resulting matrix.

**Observation** **3.**
*The rows of $A^{(9),1}$ are zero.*


**Observation** **4.**
*$A^{(9),3}$ maps to $\sum_{b=1}^{{\left(0,1\right)}_{Z_{m}}-1}T_{b,0}\left[R\right]+\sum_{j=0}^{p_{1}-2}R_{0,1+j\cdot {\left(1,0\right)}_{Z_{m}}}\left[R\right]$.*


Observation 3 follows easily from the form of the matrix $A^{\left(8\right)}$ and Lemmas 2 and 3, which implies that $\mathrm{span}(Rows\left(T_{0}\right))$ is exactly the kernel of *S* from Lemma 2 (this is the reason we need Lemma 2: it is easier to verify that a given vector is not in Ker(span(S)), rather than verifying it is not in $span(T_{0})$).

Observation 4 follows by the structure of $A^{\left(8\right)}$ and definition of *L*.

Now, if $A^{(8),3}$ is spanned by the rest of the rows in $A^{\left(8\right)}$, then it must be the case that the same dependence exists in $A^{\left(9\right)}$. Thus, it suffices to prove that the latter does not hold. Assume for the sake of contradiction that:$$v\left(A^{(9),1};A^{(9),2}\right)=A^{(9),3}$$
for some vector *v*. In the following, we use $V^{0}$ for viewing *v* as a block vector with Level-0 blocks. Note that unusually for this type of matrix, the blocks in the first row have $p_{2}-1$ rows, and in other block row, the cells have *m* rows, as usual. Similarly, we use $V^{1}$ to impose Level-1 structure onto *v*.

By the structure of $R_{1}^{2,-}$, we conclude that $A^{(9),2}$ is of the form:

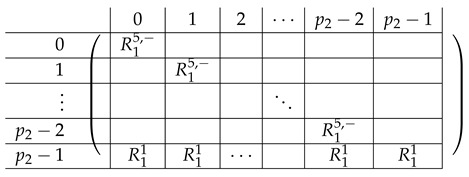


Observe that in $A^{(9),3}$, non-zero values exist only in Level-1 blocks i=0,1. As there are $p_{2}$ such blocks and $p_{2}>2$, by our assumption, we conclude that the last row contributes zero to $v(A^{(9),1};A^{(9),2})$, as in the last block, the output needs to be zero, and it equals $V\left[p_{2}\right]A^{(9),2}$, which is the same contribution for all (Level-1) blocks.

To agree with $A^{(9),3}$ at block *i*, we must then have $v\cdot R_{1}^{5,-}=e_{0}$. Viewing $R_{1}^{5,-}$ as a block-matrix of Level00, because of the zeroes at all blocks but Block 0, the contributions of all block-rows but the first one to $v\cdot R_{1}^{5,-}$ is:$$-\left(2+\left(p_{1}-2\right)\right)\cdot V^{1}\left[p_{2}\right]=-p_{1}\cdot V^{1}\left[0\right]=0.$$

In the above, the last equality is due to the fact that $p=p_{1}$. Thus, we must have $V^{1}\left[0\right]\cdot C=e_{0}$. However, we observe that Rows(C) is a subset of Ker(span(S)), where *S* is specified in Lemma 3 and thus cannot equal $e_{0}$ (which is not in Ker(span(S))).

This concludes the proof of Theorem 4.

## 5. Future Directions

Our work leaves several interesting problems open.
For what other parameters is a share conversion from (2,3)-CNF to three-additive possible? For instance, surprisingly, $p\neq p_{1},p_{2}$ is possible for (m=7·73,p=2) as follows from [10]. Our analysis does not explain this phenomenon, as we did not complete the full analysis of the resulting matrix. We believe that given the work we have done, this is a very realistic goal.We need to understand share conversion for different sets *S*. One direction is by considering *m*’s, which are a product of more than two primes. As discussed in the Introduction, already using three primes, a conversion from (2,3)-CNF over $S_{m}$ would improve over the best known constructions for three-server PIR via the BIKO paradigm. One advantage of such schemes following the BIKO framework is the constant answer size achieved. Here, we initially worked with two primes, rather than with three, to develop the tools and intuition for a slightly technically simpler setting.As discussed in the Introduction, some of the previous results not falling in the BIKO framework can be viewed as instances of an extended BIKO framework using a “many-to-one” share conversion. Viewing PIR protocols as based on share conversions between secret sharing schemes is apparently “the right” way to look at it. As certain evidence, using the more redundant CNF instead of Shamir as in [10] was a useful insight from secret sharing allowing us to further improve CC. In particular, in the case that there are no share conversions for $C_{S_{m}}$ for *m*’s that are a product of three or more primes, perhaps a suitable many-to-one conversion may still exist.Further extending this view could lead to new insights on PIR design. In particular, in many of the existing schemes, a shallow circuit is evaluated essentially by performing share conversions. In particular, local evaluation of linear functions over the inputs is a special case of such a many-to-one conversion (from a linear scheme to itself). In all schemes we have surveyed here, PIR for a large family F of functions (of size $2^{2^{n}}$) was implemented via share conversion for a small set of relations. For instance, BIKO’s 3-PIR was based on $2^{n}$ linear functions (implementing inner products with vectors in the MV family) and a share conversion for $C_{S_{m}}$, thus only $O(2^{n})$ relations. Similarly, in the two-server PIR, the situation is similar.One concrete direction may be proving lower bounds on PIR protocols based on “circuits” containing only certain share conversion “gates”. Perhaps analogies to circuit complexity could be made, borrowing techniques from circuit lower bounds. Insights for such limited classes of schemes could hopefully advance our understanding of lower bounds for PIR CC; currently, the best-known lower bound even for two-server PIR is 5n, only slightly above trivial [15].

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
