# Peer review of "On Share Conversions for Private Information Retrieval"

_entropy, 2019, doi:10.3390/e21090826_

Round 1

Reviewer 1 Report

This article is difficult to place in context.  The title refers to PIR -- there is a rule when writing that you should always spell out acronyms or abbreviations the first time they are used.  PIR is used in the title and abstract but is not defined until the first sentence in the text.  “Share conversion” is similarly not defined (and then weakly) until the body of the text. 

In the abstract, the authors claim that they are proposing a modification of what is in citation number 1, but this citation is to a 5 year old unpublished work, making it difficult for most specialists in the area to understand the arguments and context in this article.   

Author Response

*Comment 1: This article is difficult to place in context. 

I have rewritten the introduction quite a bit, to provide more context and motivation to this work.

*Comment 2:

The title refers to PIR -- there is a rule when writing that you should always spell out acronyms or abbreviations the first time they are used.  PIR is used in the title and abstract but is not defined until the first sentence in the text. 

Reply: Fixed - spelled out in title an abstract.

*Comment 3: “Share conversion” is similarly not defined (and then weakly) until the body of the text. 

This is a fairly well-known primitive, and explaining it in the abstract, would be distracting, I am afraid (please suggest otherwise, if you disagree).

*Comment 4: In the abstract, the authors claim that they are proposing a modification of what is in citation number 1, but this citation is to a 5 year old unpublished work, making it difficult for most specialists in the area to understand the arguments and context in this article.   

The work has been published, but indeed, the reference mistakenly did not include a reference of the proceedings. There is indeed one theorem in the body if the technical parts we include, referring to [1] (stated proved there), which appears in an unpublished full version but not in the published conference version. If you want to verify the theorem indeed appears there as stated, I already asked one of the authors to put the full version on their homepage, hope to get a reply soon (also, I believe they would provide it upon request for the sake of this particular review).

Reviewer 2 Report

This work builds heavily upon a previous work. This is fine, but to be more friendly to future readers, the introduction should contain some justification on why this problem is interesting. Currently too many notions are directly given (e.g., multiparty computation, secure sharing). Could help to add more details.

PIR protocols built upon the share conversion problem proposed in [1] were already beaten in 2 and 3 servers settings. So I am wondering if the newer and more efficient PIR protocols also follow from the share conversion paradigm? In general, as the share conversion problem is mainly used for PIR, I would like to see more discussions on this (e.g., more PIR protocols produced? current best ones recovered?).

Author Response

Thanks you very much for your comments. Following are some replies.

*Comment 1:

This work builds heavily upon a previous work. This is fine, but to be more friendly to future readers, the introduction should contain some justification on why this problem is interesting. Currently too many notions are directly given (e.g., multiparty computation, secure sharing). Could help to add more details.

Reply:

We have rewritten the introduction quite a bit, to provide more context and motivation to this work. We also added a self-contained presentation of the relevant secret sharing related definitions and constrictions. The application to MPC of PIR is somewhat far from the focus of the paper, providing some context for why PIR is important. So we did not elaborate on this connection further. Perhaps it is best to consider PIR on its own right. It is a non-trivial primitive allowing to achieve small communication complexity of querying a replicated database, while keeping the index of the client private.

Do you suggest to discuss applications of PIR itself, rather than being an instance of a larger setting of secure multi party computation?

*Comment 2:

PIR protocols built upon the share conversion problem proposed in [1] were already beaten in 2 and 3 servers settings. So I am wondering if the newer and more efficient PIR protocols also follow from the share conversion paradigm? In general, as the share conversion problem is mainly used for PIR, I would like to see more discussions on this (e.g., more PIR protocols produced? current best ones recovered?).

Reply: We rewrote the intro to include a more detailed discussion of how the best known PIR protocols fit into the share conversion paradigm. In a nutshell, Dvir and Gopi (I believe this is the work you are referring to) for 2-server PIR do not fit into the BIKO framework, but rather into an extended BIKO framework (see "previous work" section). In particular, the share domains of the secret sharing schemes involved are not constant in DG14, and are constant in BIKO. They do not explicitly provide  an improved 3-server PIR. I would guess that using order-2 derivatives would allow to improve by using S_m-MV-families, for m which is a product of 3 primes (and S_m is of size 2^3 = 8), but I am not sure, and they do not explicitly state it.

Is this what you meant?  If this indeed works, then this construction indeed beats BIKO's construction. Will fix the writeup in this case (or if any other improvement for 3-server PIR exists that I am not aware of). 

Round 2

Reviewer 2 Report

The revision looks good. The author could consider adding some responses to the main document if proper. A minor comment is that the reference in the first page is missing.